# Bacterial degradation activity in the Eastern Tropical South Pacific oxygen minimum zone

Marie Maßmig, Jan Lüdke, Gerd Krahmann, Anja Engel*

GEOMAR Helmholtz Centre for Ocean Research Kiel, Düsternbrooker Weg 20, D-24105 Kiel, Germany

*Correspondence to*: Anja Engel (aengel@geomar.de)

**Abstract.** Oxygen minimum zones (OMZs) show distinct biogeochemical processes that relate to microorganisms being able to thrive under low or even absent oxygen. Microbial degradation of organic matter is expected to be reduced in OMZs, although quantitative evidence is low. Here, we present heterotrophic bacterial production ($^3$H leucine-incorporation), extracellular enzyme rates (leucine aminopeptidase /$ß$-glucosidase) and bacterial cell abundance for various *in situ* oxygen concentrations in the water column, including the upper and lower oxycline, of the Eastern Tropical South Pacific off Peru. Bacterial heterotrophic activity in the suboxic core of the OMZ (at *in situ* $\leq 5$ µmol $O_2$ kg$^{-1}$) ranged from 0.3 to 281 µmol C m$^{-3}$ d$^{-1}$ and was not significantly lower than in waters of 5-60 µmol $O_2$ kg$^{-1}$. Moreover, bacterial abundance in the OMZ and leucine aminopeptidase activity were significantly higher in suboxic waters compared to waters of 5-60 µmol $O_2$ kg$^{-1}$, suggesting no impairment of bacterial organic matter degradation in the core of the OMZ. Nevertheless, high cell-specific bacterial production was observed in samples from oxyclines and cell-specific extracellular enzyme rates were especially high at the lower oxycline, corroborating earlier findings of highly active and distinct micro-aerobic bacterial communities. To assess the impact of bacterial degradation of dissolved organic matter (DOM) for oxygen loss in the Peruvian OMZ, we compared diapycnal fluxes of oxygen and dissolved organic carbon (DOC) and their microbial uptake within the upper 60 m of the water column. Our data indicate low bacterial growth efficiencies of 1-21% at the upper oxycline, resulting in a high bacterial oxygen demand that can explain up to 33% of the observed average oxygen loss over depth. Our study therewith shows that microbial degradation of DOM has a considerable share in sustaining the OMZ off Peru.

## 1. Introduction

In upwelling zones at eastern continental margins, oxygen minimum zones (OMZs) with hypoxic (<60 µmol $O_2$ kg$^{-1}$), suboxic (<5 µmol $O_2$ kg$^{-1}$) or even anoxic conditions occur (Gruber, 2011; Thamdrup et al., 2012; Tiano et al., 2014). OMZs have expanded over the past years resulting in an ~3.7 % increase of hypoxic waters at depth (200 dbar) between 1960 and 2008 (Stramma et al., 2010). One of the largest anoxic water masses in the global ocean (2.4 x $10^{13}$ m$^3$) is located in the Eastern Tropical South Pacific and includes the Peruvian upwelling system (Kämpf and Chapman, 2016; Paulmier and Ruiz-Pino, 2009; Thamdrup et al., 2012). There, nutrient-rich water is upwelled and supports high rates of primary production and accumulation of organic matter. Biological degradation of organic matter subsequently reduces oxygen below the surface mixed layer (Kämpf and Chapman, 2016). As a consequence, and supported by sluggish ventilation of water masses, a permanent OMZ forms between 100 and 500 m depth, with upper and lower boundaries, i.e. oxyclines, varying within seasonal and inter-annual cycles (Czeschel et al., 2011; Graco et al., 2017; Kämpf and Chapman, 2016). In austral winter, upwelling and subsequently the nutrient supply to the surface waters increase (Bakund and Nelson, 1991; Echevin et al., 2008). However, chlorophyll *a* (Chl *a*) concentration is highest in austral summer, with the seasonal amplitude being stronger for surface than for depth averaged Chl *a* concentrations (Echevin et al., 2008). In winter, phytoplankton growth is, next to iron, mainly limited by light due to the deeper mixing, whereas in summer macronutrients can become a limiting factor (Echevin et al., 2008). Further, El Niño–Southern Oscillation may affect organic matter cycling in the area since it affects the depth of the oxycline and therefore the extent of anaerobic processes in the upper water column (Llanillo et al., 2013). During the year of this study (2017), neither a strong La Niña nor a strong El Niño was detected (https://ggweather.com/enso/oni.htm). However, in January, February and March 2017 there was a strong coastal El Niño with enhanced warming (+1.5°C) of sea surface temperatures in the eastern Pacific (Garreaud, 2018).

Within OMZs, enhanced vertical carbon export has been observed (Devol and Hartnett, 2001; Roullier et al., 2014) and explained by a potentially reduced remineralization of organic matter in suboxic and anoxic waters. This is possibly because microbes apply anaerobic respiratory pathways that yield less metabolic energy compared to aerobic respiration. For instance, denitrification or dissimilatory nitrate reduction to ammonia (DNRA) result only in 99 %, or 64 % of the energy (kJ) per oxidized carbon atom that is produced by aerobic respiration (Lam and Kuypers, 2011). Additionally, the energy yield available for the production of cell mass seems to be less than expected from the chemical equations (Strohm et al., 2007). Meanwhile, bacteria are mainly responsible for the remineralization of organic matter into nutrients and carbon dioxide ($CO_2$) in the ocean (Azam et al., 1983). Thus, microbial activity and consequently organic matter remineralization in suboxic and anoxic waters might be reduced, possibly explaining enhanced export of carbon. As a consequence, expanding OMZs could result in increased $CO_2$ storage in the ocean.

During the degradation process, low molecular weight (LMW <1 kDa) organic compounds can directly be taken up by bacteria (Azam et al., 1983; Weiss et al., 1991). However, in the ocean, bioavailable organic matter is commonly in the form of particulate organic matter or high molecular weight (HMW) DOM (Benner and Amon, 2015). To access this organic matter pool, bacteria produce extracellular, substrate specific enzymes that hydrolyse polymers

into LMW units (Hoppe et al., 2002). Taken-up, organic matter is partly incorporated into bacterial biomass, or
respired to $CO_2$, which may evade to the atmosphere (Azam et al., 1983). Rates of enzymatic organic matter
hydrolysis or bacterial production are controlled by the environment, i.e. temperature and pH, but can be actively
regulated e.g. in response to changing organic matter supply and quality (Boetius and Lochte, 1996; Grossart et al.,
2006; Pantoja et al., 2009; Piontek et al., 2014). However, the effect of oxygen concentration, which dictates the
respiratory pathway and thus energy gain, on bacterial production and the expression of extracellular enzymes in
aquatic systems, is poorly understood. For instance, bacterial production was higher in anoxic lake waters (Cole and
Pace, 1995), whereas in the Pacific waters off Chile bacterial production and DOM decomposition rates did not
change in relation to oxygen concentrations (Lee, 1992; Pantoja et al., 2009). Investigations of hydrolysis rates as the
initial step of organic matter degradation, may help to unravel possible adaptation strategies of bacterial communities
to suboxic and anoxic conditions (Hoppe et al., 2002). High extracellular enzyme rates might compensate a putative
lower energy yield of anaerobic respiration and the subsequent biogeochemical effects. However, very few studies
have investigated the effect of oxygen on hydrolytic rates, so far. Hoppe et al. (1990) did not find differences
between oxic and anoxic incubations of Baltic Sea water. In the Cariaco Basin, hydrolytic rates were significantly
higher in oxic compared to anoxic water (Taylor et al., 2009). However, this difference did not persist after rates
were normalized to particulate organic matter concentration. The dependence of hydrolysis rates on organic matter
concentrations described by Taylor et al. (2009), suggest that productivity may play a role for extracellular
enzymatic rates in oxygen depleted systems. The Peruvian upwelling system displays high amounts of labile organic
matter (Loginova et al., 2019) at shallow oxyclines and thus allows for studying effects of low oxygen on
extracellular enzyme rates under substrate replete conditions. In general, combined investigations of extracellular
enzyme rates, bacterial production (measured by [3]H leucine-incorporation) and carbon fluxes sampled at various *in*
*situ* oxygen concentrations are still missing. These data, however, are crucial to inform ocean biogeochemical
models that aim at quantification of $CO_2$ uptake and nitrogen loss processes in oxygen depleted areas.
We studied bacterial degradation of organic matter in the OMZ off Peru during an extensive sampling campaign in
the Austral winter 2017. We determined rates of total and cell-specific bacterial production ([3]H leucine-
incorporation) as well as of leucine aminopeptidase (LAPase) and *β*-glucosidase (GLUCase). We estimate bacterial
utilisation of DOC supplied by diapycnal transport into the OMZ and discuss the contribution of bacterial
degradation activity to the formation and persistence of the OMZ off Peru.
## 2. Methods
### 2.1. Study site and CTD measurements
Samples were taken during the cruises M136 and M138 on the R/V METEOR off Peru in April and June 2017,
respectively (Fig. 1). Seawater was sampled with 24 Niskin bottles (10 L) on a general oceanic rosette system. At
each station, 5 to 11 depths were sampled between 3 and 800 m (supplementary Table 1). Oxygen concentrations,
temperature and depth were measured with a Sea-Bird SBE 9-plus CTD System (Sea-Bird Electronics, Inc., USA).
Oxygen concentrations at each depth were determined with a SBE 43 oxygen sensor, calibrated with Winkler
titrations (Winkler, 1888), resulting in an overall accuracy of 1.5 µmol kg$^{-1}$ oxygen. Chl *a* fluorescence was detected
with a WETStar Chl *a* sensor (WET Labs, USA) and converted to µg l$^{-1}$ using factors given by the manufacturer
(Wetlabs).

### 2.2. Dissolved organic carbon, total dissolved nitrogen, dissolved hydrolysable amino acids and dissolved high molecular weight carbohydrates

DOC and total dissolved nitrogen (TDN) samples were taken at all stations, whereas the further analysis of DOC
data was limited to stations with compatible bacterial production data and turbulence measurements (stations G-T).
For DOC and TDN 20 ml of seawater was sampled in replicates, whereas both replicates were only analysed in case
of conspicuous data. Samples were filtered through a syringe filter (0.45 µm glass microfiber GD/X membrane,
Whatman ™) that was rinsed with 50 ml sample, into a combusted glass ampoule (8 h, 500 °C). Before sealing the
ampules, 20 µl of 30 % ultrapure hydrochloric acid were added. Samples were stored at 4 °C in the dark for 3 months
until analyses. DOC and TDN were analysed using a TOC−VCSH with a TNM-1 detector (Shimadzu), applying a
high-temperature catalytic oxidation method modified from Sugimura and Suzuki (1988). The instrument was
calibrated with potassium hydrogen phthalate standard solutions (0 to 416.7 µmol C l$^{-1}$) (Merck 109017) and a
potassium nitrate standard solution (0-57.1 µmol N l$^{-1}$) (Merck 105065). The instrument blank was examined with
reference seawater standards (Hansell laboratory RSMAS University of Miami). The relative standard deviation
(RSD) between repeated measurements is <1.1 % and <3.6 % and the detection limit is 1 µmol l$^{-1}$ and 2 µmol l$^{-1}$ for
DOC and TDN, respectively.
At each station replicate 4 ml and 16 ml sample for the analysis of dissolved amino acids (DHAA) and dissolved
combined carbohydrates (DCHO) were filtered through rinsed Acrodisc® 0.45µm GHP membrane (Pall) and stored
in combusted vials (8 h, 500 °C) at -20 °C, respectively. Replicates were only analysed, if the first sample analyses
resulted conspicuous data. The following DHAA were analysed: Alanine, Arginine, Glycine, Leucine,
Phenylalanine, Serine, Threonine, Tyrosine, Valine, Aspartic acid + Asparagine (co-eluted), Glutamine + Glutamic
acid (co-eluted), γ-Aminobutyric acid and Isoleucine. DHAA samples were analysed with a high performance liquid
chromatograph (1260 HPLC system, Aglient Technologies) using a C$_{18}$ column (Phenomex Kinetex) after in line
ortho-phthaldialdehyde derivatization with mercaptoethanol after Lindroth and Mopper (1979) and Dittmar et al.
(2009) with slight modifications after Engel and Galgani (2016). DCHO samples were desalted by membrane
dialysis (1kDa, Spectra Por) and analysed with a high performance anion exchange chromatography (HPAEC)
(DIONEX ICS3000DC) after Engel and Händel (2011). Detection limit of DHAA was 1.4 nmol L$^{-1}$ depending on
amino acid and 10 nmol L$^{-1}$ for DCHO. The precision was 2% and 5% for DHAA and DCHO, respectively.

### 2.3. Diapycnal fluxes of oxygen and dissolved organic carbon

In this study, we calculated DOC and oxygen loss rates (mmol m$^{-3}$ d$^{-1}$) from the changes in diapycnal fluxes over
depth. Therefore, oxygen and DOC profiles were used (stations G-T), excluding the mixed layer, defined by
temperature deviating ≤0.2°C from the maximum, but excluding at least the upper 10 m. The diapycnal flux ($\Phi_S$)
was calculated for each CTD profile (Fischer et al., 2013; Schafstall et al., 2010) assuming a constant gradient
between two sampled depths for DOC and oxygen:
1.  $\Phi_S = -K_\rho \nabla C_S$

where $\nabla C_S$ is the gradient (mol m$^{-4}$). The diapycnal diffusivity of mass ($K\rho$) (m$^2$ s$^{-1}$) was assumed to be constant ($10^{-3}\ m^2 s^{-1}$ ), which is reasonable compared with turbulence measurements by a freefalling microstructure probe (see supplementary methods and Fig. 2a). DOC loss rates ($\nabla\Phi_{DOC}$; mmol m$^{-3}$ d$^{-1}$) and oxygen loss rates ($\nabla\Phi_{DO}$; mmol m$^{-3}$ d$^{-1}$) were assumed to be equal to the negative vertical divergence of $\Phi s$ calculated from the mean diapycnal flux profile, implying all other physical supply processes to be negligible.

## 2.4. Bacterial abundance

Bacterial abundance was sampled in replicates at each station, whereas replicates were only analysed in exceptions. Abundance was determined by flow cytometry after Gasol and Del Giorgio (2000) from 1.6 ml sample, fixed with 0.75µl 25 % glutaraldehyde on board and stored at -80°C for maximal 3 month until analyses. Prior to analysis samples were thawed and 10 µL Flouresbrite® fluorescent beads (Polyscience, Inc.) and 10 µL Sybr Green (Invitrogen) (final concentration: 1x of the 1000x Sybr Green concentrate) were added to 400 µl sample. Cells were counted on a FACS Calibur (Becton Dickinson), calibrated with TruCount Beads ™ (BD) with a measurement error of 2 % RSD.

## 2.5. Bacterial production, oxygen demand and growth efficiency

For bacterial production, the incorporation of radioactive labelled leucine ($^3$H) (specific activity 100 Ci mmol$^{-1}$, Biotrend) was measured (Kirchman et al., 1985; Smith and Azam, 1992) at all depths of stations G-T as replicates. For this, the radiotracer at a saturating final concentration of 20 nmol l$^{-1}$ was added to 1.5 ml of sample and incubated for 3 hours in the dark at 13°C. Controls were poisoned with trichloracetic acid. Samples were measured with a liquid scintillation counter (Hidex 300 SL, TriathalerTM, FCI). Samples taken at *in situ* oxygen concentrations of < 5 µmol kg$^{-1}$ were incubated under anoxic conditions by gentle bubbling with gas (0.13 % CO$_2$ in pure N$_2$). Samples from oxic waters were incubated with head space, without bubbling. All samples were shacked thoroughly in between, therefore the bubbling of just one treatment won't have any effect. $^3$H-leucine uptake was converted to carbon units applying a conversion factor of 1.5 kg C mol$^{-1}$ leucine (Simon and Azam, 1989). An analytical error of 5.2 % RSD was estimated with triplicate calibrations. Samples with a SD (standard deviation) > 30% between replicates were excluded.

The incubation of samples at a constant temperature of 13°C resulted in deviations of max. 11°C between incubation ($T_{incubation}$) and *in situ* temperatures ($T_{insitu}$). In order to estimate *in situ* bacterial production from measured bacterial production during incubations, measured temperature differences were taken into account following the approach of López-Urrutia and Morán (2007). First, the temperature difference between $T_{insitu}$ and $T_{incubation}$ ($\delta T$) was computed in electron volt (ev$^{-1}$), after $T_{insitu}$ and $T_{incubation}$ (K) had been multiplied with the Boltzmann's constant $k$ *(8.62x10$^{-5}$ eV K$^{-1}$):*

2.  $\delta T\ [ev^{-1}] = \dfrac{1}{T_{incubation}[K]\ x\ k\ [evK^{-1}]} - \dfrac{1}{T_{insitu}[K]\ x\ k\ [evK^{-1}]}$

The decadal logarithm of *in situ* bacterial production ($\log_{10} BP_{insitu}$) was then calculated from the decadal logarithm of
measured bacterial production during incubations ($\log_{10} BP_{incubation}$). Therefore we applied three different factors (*F*)
depending on *in situ* Chl *a* concentration as proposed by López-Urrutia and Morán (2007); with *F* being -0.583, -0.5
and -0.42 $[fgCcell^{-1}d^{-1}ev]$ for <0.5, 0.5-2 and >2 µg Chl *a* L$^{-1}$, respectively:

168        3.    $log_{10}BP_{insitu}[fgCcell^{-1}d^{-1}] =$

$$log_{10}BP_{incubation}[fgCcell^{-1}d^{-1}] + \delta T\,[ev^{-1}]x\,F\,[fgCcell^{-1}d^{-1}ev]$$

Within the text, figures, equations and statistic results it is always referred to temperature corrected *in situ* bacterial
production. Temperature corrected bacterial production and original bacterial production measured during incubation
can be compared in supplementary Table 2.
The bacterial oxygen demand (BOD; mmol O$_2$ m$^{-3}$ d$^{-1}$) is the amount of oxygen needed to fully oxygenize organic
carbon that has been taken up and not transformed into biomass by bacterial production (mmol C m$^{-3}$ d$^{-1}$). The BOD
was calculated as the difference between the estimated bacterial DOC uptake and the bacterial production applying a
respiratory quotient (*cf*) of 1 (Eq. (4)) (Del Giorgio and Cole, 1998).

176        4.    $BOD = (DOC\ uptake - bacterial\ production) \times cf$

The bacterial DOC uptake was calculated under two different assumptions: i) the DOC uptake by bacteria equals the
DOC loss rate over depth or ii) the bacterial growth efficiency (BGE) follows the established temperature
dependence (BGE=0.374[±0.04] -0.0104[±0.002]T [°C]), resulting in a BGE between 0.1 and 0.3 in the depth range
of 10-60 m and an *in situ* temperature of 14 to 19°C (Rivkin and Legendre, 2001) and can be used to estimate the
bacterial DOC uptake from bacterial production (Eq. (5)).

182        5.    $bacterial\ DOC\ uptake = \frac{bacterial\ production}{BGE}$

**2.6. Extracellular enzyme rates**
Potential hydrolytic rates of LAPase and GLUCase were determined with fluorescent substrate analogues (Hoppe,
1983). L-leucine-7-amido-4-methylcoumarin (Sigma Aldrich) and 4-methylumbelliferyl-ß-D-glucopyranoside
(Acros Organics) were added in final concentrations of 1, 5, 10, 20, 50, 80, 100 and 200 µmol l$^{-1}$ in black 69 well
plates (Costar) and kept frozen for at most one day until replicates of 200 µl sample were added. After 0 and 12
hours of incubation at 13°C in the dark, fluorescence was measured with a plate reader fluorometer (FLUOstar
Optima, BMG labtech) (excitation: 355 nm; emission: 460 nm). An error of 2 % RSD was defined using the
calibration with triplicates. Blanks with MilliQ were performed to exclude an increase in substrate decay over time.
Samples were collected in replicates (*n*=2) at station A-K and incubated directly after sampling under oxygen
conditions resembling *in situ* oxygen conditions. For samples > 5 µmol *in situ* O$_2$ kg$^{-1}$ incubations were conducted
under atmospheric oxygen conditions. Samples < 5 µmol *in situ* O$_2$ kg$^{-1}$ were incubated in a gas tight incubator that
had two openings to fill and flush it with gas. For our experiment the incubator was flushed and filled with $N_2$, to
reduce oxygen concentrations. Still control measurements occasionally revealed oxygen concentrations of 8 to 40
$\mu$mol $O_2$ kg$^{-1}$. Additionally, samples were in contact with oxygen during pipetting and measurement. To investigate
the influence of the different incubation methods we additionally incubated samples > 5 $\mu$mol *in situ* $O_2$ kg$^{-1}$ under
reduced oxygen concentrations. On average incubations under reduced oxygen concentration yielded 2-27% higher
values than those incubated under atmospheric oxygen conditions. However, the observed trends over depth
remained similar (see supplementary discussion).
Calibration was conducted with 7-amino-4-methylcoumarin (2 nmol l$^{-1}$ to 1 $\mu$mol l$^{-1}$) (Sigma Aldrich) and 4-
methylumbelliferone (Sigma Aldrich) (16 nmol l$^{-1}$ to 1 $\mu$mol l$^{-1}$) in seawater at atmospheric oxygen concentrations
and under $N_2$ atmosphere.
Maximum reaction velocity ($V_{max}$) at saturating substrate concentrations was calculated using both replicates at once,
with the simple ligand binding function in SigmaPlot™ 12.0 (Systat Software Inc., San Jose, CA). Values for $V_{max}$
with a SD >30 % were excluded from further analyses. The degradation rate ($\delta$) [$\mu$mol C m$^{-3}$ d$^{-1}$] of DHAA by
LAPase and DCHO by GLUCase was calculated after Piontek et al. (2014):
6.   $\delta = \frac{h_r * c}{100}$
where $h_r$ [% d$^{-1}$] is the hydrolyses turnover at $10^3$ $\mu$mol m$^{-3}$ substrate concentration and $c$ is the carbon content of
DHAA [$\mu$mol C m$^{-3}$]. Measurements of $h_r$ with a SD between duplicates of more than 30% were excluded. The same
procedure was conducted with the carbon content of dissolved hydrolysable leucine, instead of DHAA, to account
for variations in leucine concentrations, which is the main amino acid hydrolysed by LAPase.
Similar to bacterial production, *in situ* extracellular enzyme rates were estimated based on extracellular enzyme rates
measured during incubation. To account for the differences between *in situ* and incubation temperatures a correction
factor (*F*) was applied based on differences in extracellular enzyme rates after additional incubations at 22.4°C next
to the regular incubations at 13°C at five stations during the cruises. The fluorescence signals at different substrate
concentrations increased on average by a factor of 0.05 and 0.03 ($°C^{-1}$) for GLUCase and LAPase, respectively.
Under the assumption that the increase in rates with temperature was linear, measured enzyme rates were adapted to
*in situ* temperature, with (*EER$_{insitu}$*; nmol L$^{-1}$ h$^{-1}$) and (*EER$_{incubation}$*) being the *in situ* extracellular enzyme rates and
extracellular enzyme rates during incubation, respectively:
7.   $\delta T \; [°C] = T_{insitu}[°C] - T_{incubation} \; [°C]$

8.   $EER_{insitu}[nmolL^{-1}h^{-1}] =$

$$EER_{incubation}[nmolL^{-1}h^{-1}] + EER_{incubation}[nmolL^{-1}h^{-1}] \; x \; F \; [°C^{-1}] \; x \; \delta T \; [°C]$$

Within the text, figures, equations and statistic results it is always referred to the temperature corrected *in situ*
extracellular enzyme rates. Temperature corrected extracellular enzyme rates and original extracellular enzyme rates
measured during incubation can be compared in supplementary Table 2.
**2.7. Data analyses**
Data were plotted with Ocean Data View 4.74 (Schlitzer, 2016), MATLAB (8.3.0.532 (R2014a)) and R version 3.4.2
using the package *ggplot2* (Hadley Wickham, 2016; R Development Core Team, 2008). Statistical significances
between different regimes (see supplementary Table 2 for mean and SD within different regimes and statistical
results) were tested with a *Wilcoxon test* (W) and correlation with the *Spearman Rank correlation* (S) in R version
3.4.2 (R Development Core Team, 2008) using following R packages: *FSA, car and multcomp* (Derek H. Olge,
2018; Horthorn et al., 2008; John Fox and Sanford Weisberg, 2011). For this extracellular enzyme data of station A-
K and bacterial production data of station G-T were used, since not all parameters could be sampled at all depth.
Diapycnal fluxes of DOC and oxygen were calculated with MATLAB (8.3.0.532 (R2014a)) and the Toolbox Gibbs
SeaWater (GSW) Oceanographic Toolbox (3.05) (McDougall and Barker, 2011).
Samples were categorized into different oxygen regimes. Due to sensitivities of oxygen measurements, we did not
distinguish between anoxic and suboxic regimes, but defined the suboxic "OMZ" oxygen regime by a threshold $\leq 5$
$\mu$mol $O_2$ kg$^{-1}$ (Gruber, 2011). We defined the oxycline as one regime (>5 to <60 $\mu$mol $O_2$ kg$^{-1}$) including the upper
and lower oxycline or separated it into "low_hypoxic" (>5 to <20 $\mu$mol $O_2$ kg$^{-1}$) and "high_hypoxic" (>20 to <60
$\mu$mol $O_2$ kg$^{-1}$) regimes, representing important thresholds of oxygen concentrations for biological processes (Gruber,
2011). Oxygen concentrations >60 $\mu$mol $O_2$ kg$^{-1}$ were defined as "oxic". Moreover, we partly differentiated between
oxygen regimes situated above and below the OMZ (see supplementary Table 2 for results).
**3.  Results**
**3.1. Biogeochemistry of the Peruvian OMZ**
During our two cruises to the Peruvian upwelling system (Fig. 1), maximum Chl *a* concentration was higher and
temperatures were warmer in April compared to June 2017, probably representing seasonal variability. Chl *a*
concentration reached up to 11 and 4 µg l$^{-1}$ within the upper 25 m in April and June, respectively. Still, average Chl *a*
concentration at depth <10 m (M136: 3.1$\pm$2.6 µg l$^{-1}$; M138: 2.8 $\pm$ 1.3 µg l$^{-1}$) were not significantly different between
the two cruises. At depths >50 m, Chl *a* concentration was generally below detection limit (Fig. 3a, supplementary
Fig. 1). At depth < 10 m the water was warmer in April (21.3 $\pm$1.6°C) than in June (17.6 $\pm$ 0.6°C) (Fig. 3b,
supplementary Fig. 1). Oxygen concentration >100 µmol kg$^{-1}$ was observed in the surface mixed layer. Oxygen
decreased steeply with depth, reached suboxic concentrations (<5 µmol kg$^{-1}$) at > 60 $\pm$ 24 m (Fig. 2c, 4a and 5a,
supplementary Fig.1) and fell below detection of Winkler titration. For further analysis and within the text *in situ*
oxygen concentrations <5 µmol $O_2$ kg$^{-1}$ are referred to as "suboxic". Shallowest depth with suboxic oxygen
concentrations was 14 m in April (station Q) and 29 m in June (station D), probably representing that station Q was
situated closer to the shore than station D. Oxygen increased again to up to 15 µmol kg$^{-1}$ at >500 m (Fig. 4a and 5a,
supplementary Fig. 1). TDN concentrations increased with depth from $18\pm8$ µmol $l^{-1}$ and $22\pm7$ µmol $l^{-1}$ within the
upper 20 m in April and June, respectively, and reached a maximum of 54 µmol $l^{-1}$ at 850 m (Fig. 3c). DOC
decreased with depth from $94 \pm37$ µmol $l^{-1}$ and $69 \pm12$ µmol $l^{-1}$ in the upper 20 m in April and June, respectively, to
lowest values of 37 µmol $l^{-1}$ at 850 m. The steepest gradient in DOC concentration was observed in the upper 20-60
m (Fig. 2b and 3d) during both cruises.

### 3.2. Bacterial production and enzymatic activity

Bacterial production varied strongly throughout the study region and ranged from 0.2 to 2404 µmol C $m^{-3}$ $d^{-1}$ (Fig.
4b), decreased in general from surface to depth (except for the most coastal station) and showed significantly higher
rates in the oxygenated surface compared to the OMZ (Fig. 4b). At the most coastal station (G) bacterial production
remained high near the bottom depth of 75 m (280 µmol C $m^{-3}$ $d^{-1}$ at 72 m) (Fig. 4b). Bacterial production did not
differ significantly between the oxyclines and the suboxic core waters, neither off-shore (suboxic: 0.3-127 µmol C $m^{-3}$
$d^{-1}$ ; oxyclines: 1-304 µmol C $m^{-3}$ $d^{-1}$) nor at the most coastal stations (G and T) (suboxic: 146-281 µmol C $m^{-3}$ $d^{-1}$)
(oxycline: 74-452 µmol C $m^{-3}$ $d^{-1}$) (see supplementary Table 2 for all statistical results). Further, no significant
correlation was observed between bacterial production and oxygen at *in situ* <20 µmol $O_2$ $kg^{-1}$. Additionally,
significantly lower bacterial production was observed within the lower oxycline (0.7-3.3 µmol C $m^{-3}$ $d^{-1}$) compared
to the core OMZ (0.3-281 µmol C $m^{-3}$ $d^{-1}$) even though oxygen increased from <5 to 15 µmol $kg^{-1}$ (Fig. 4a, b).
Trends between oxygen regimes were similar between temperature corrected bacterial production (presented
throughout the text) and original bacterial production measured during incubation (supplementary Table 2).
Overall, bacterial abundance ranged from 1 to 49 x $10^5$ cells $ml^{-1}$, with highest abundance observed at the surface and
close to the sediment. Cell abundance in the oxyclines (1-16 x $10^5$ cells $ml^{-1}$) was significantly lower than in the
OMZ core (1-25 x $10^5$ cells $ml^{-1}$) (Fig. 4c). A sharp decrease in bacterial abundance was observed below the OMZ.
Estimates for the *in situ* degradation rate of DHAA by LAPase take into account the available concentrations of
DHAA and varied between 0.7 and 39.7 µmol C $m^{-3}$ $d^{-1}$. LAPase degradation rates observed within the OMZ core
($5.5 \pm 2.1$ µmol C $m^{-3}$ $d^{-1}$) were significantly higher than in the oxyclines ($3.1 \pm 2.3$ µmol C $m^{-3}$ $d^{-1}$) (Fig. 5b). To
exclude an influence of changing DHAA composition over depth, LAPase activity was also calculated using *in situ*
concentrations of dissolved hydrolysable leucine instead of total DHAA. Degradation rates of dissolved hydrolysable
leucine by LAPase (0.01-1.92 µmol C $m^{-3}$ $d^{-1}$) showed the same trend with significantly higher rates in suboxic
waters than in the oxyclines. Thus, differences in the molecular composition of DHAA had no influence on spatial
degradation patterns being higher in suboxic waters than in the upper oxycline. In contrast, degradation rates of
DCHO (>1kDa) were slightly reduced within the suboxic waters ($0.69 \pm 1.30$ µmol C $m^{-3}$ $d^{-1}$) compared to the
oxyclines ($1.1 \pm 1.0$ µmol C $m^{-3}$ $d^{-1}$) (Fig. 5c). Since degradation rates were calculated by multiplying enzyme rates
and carbon concentrations of DCHO and DHAA at *in situ* depth, differences in carbon concentrations are important
for further interpretation. *In situ* carbon concentrations of DHAA were similar between the OMZ core ($0.53\pm 0.1$
µmol C $L^{-1}$) and the oxycline ($0.57\pm 0.2$ µmol C $L^{-1}$). In contrast, *in situ* carbon concentrations of DCHO were
reduced within the OMZ core ($1.3 \pm 0.4$ µmol C $L^{-1}$) compared to the oxycline ($1.5 \pm 0.6$ µmol C $L^{-1}$) (Fig. 3e, f),
suggesting that calculated differences between degradation rates may be influenced by different carbon
concentrations. Potential hydrolytic rates at saturating substrate concentration ($V_{max}$) of LAPase ranged between 9
and 158 nmol $l^{-1}$ $h^{-1}$ and were ~30 times lower for GLUCase. LAPase $V_{max}$ was significantly higher within the
suboxic waters (50 ± 21 nmol $l^{-1}$ $h^{-1}$) compared to the oxycline (36 ± 20 nmol $l^{-1}$ $h^{-1}$) and GLUCase $V_{max}$ was more
similar within the suboxic waters (1.6 ± 1.5 nmol $l^{-1}$ $h^{-1}$) compared to the oxycline (1.2 ± 0.6 nmol $l^{-1}$ $h^{-1}$) (Fig. 5d, e).
Trends between oxygen regimes were similar between temperature corrected extracellular enzyme rates (presented
throughout the text) and extracellular enzyme rates measured during incubation (supplementary Table 2).
To investigate physiological effects of suboxia, we normalized bacterial production and enzymatic rates to cell
abundance. Cell-specific production ranged between 1 and 1120 amol C $cell^{-1}$ $d^{-1}$ (Fig. 4d). In contrast to total
production, cell-specific production was significantly higher at the oxyclines compared to suboxic core waters at the
off-shore stations (suboxic: 1-102 µmol C $m^{-3}$ $d^{-1}$, oxyclines: 6-219 µmol C $m^{-3}$ $d^{-1}$). At the most coastal stations (G
and T) cell-specific rates were more similar between suboxic waters and the oxyclines (suboxic: 129-135 µmol C $m^{-3}$
$d^{-1}$) (oxycline: 72-284 µmol C $m^{-3}$ $d^{-1}$). Further, cell-specific bacterial production was slightly correlated (spearman
rank correlation =0.36) to oxygen concentrations at ≤20 µmol $O_2$ $kg^{-1}$ and as long as the most coastal stations (G and
T) were included this correlation was significant (Fig. 4d, supplementary Table 2). A detailed view at total- and cell-
specific bacterial production in dependence of *in-situ* oxygen concentrations, reveals a stronger increase of cell-
specific bacterial production, especially at <10 µmol $O_2$ $kg^{-1}$ at different stations (supplementary Fig. 2).
Cell-specific degradation rates of DHAA increased with depth and yielded significantly higher rates at the lower
oxycline compared to all shallower depths. Cell-specific LAPase $V_{max}$, GLUCase $V_{max}$ and GLUCase degradation
rate showed the same trends, however for the latter this trend was not significant (Fig. 5g-j, supplementary Table 2)

### 313 3.3. Bacterial contribution to the loss of dissolved organic carbon and oxygen in the
### 314 oxycline

We calculated the loss of oxygen and DOC during physical transport from below the mixed layer depth (MLD; 10-
32 m) to 60 m based on observed changes in diapycnal fluxes (Eq. (1), Fig. 2b, c). We estimated the bacterial
contribution to this loss using two different approaches (Table 1): i) We assumed that the loss of DOC over depth
equalled the bacterial uptake implying that the DOC is subsequently incorporated as bacterial biomass (bacterial
production) or respired to $CO_2$ (Eq. (4)) ii) the amount of DOC taken up by bacteria was determined by the measured
bacterial incorporation of carbon (bacterial production) and a constant ratio between carbon that is taken up and
carbon that is incorporated as biomass (bacterial production) (Eq. (5)) (see section 2.5 for details). This ratio (BGE),
was here assumed to be 10 or 30%, based on the empirical equation by Rivkin and Legendre with an *in situ*
temperature that varied between 14 and 19°C (Rivkin and Legendre, 2001).
For total average DOC loss ($\nabla\Phi_{DOC}$), we calculated a range of 1.13-3.40 mmol C $m^{-3}$ $d^{-1}$, with loss rates decreasing
most strongly below the shallow mixed layer down to 40 m (Table 1, Fig. 2c). Following the first (i) assumption, all
DOC that was lost over depth was taken up by bacteria and the measured bacterial production represents the fraction
of DOC that was incorporated as biomass. Consequently, the remaining DOC that has been taken up, in other words
the difference between DOC loss and bacterial production (0.03- 0.71 mmol C $m^{-3}$ $d^{-1}$), was respired to $CO_2$ and
represents the bacterial oxygen demand to account for the DOC loss ($BOD_\varepsilon$) (0.98-3.36 mmol $O_2$ $m^{-3}$ $d^{-1}$) (Eq. (4)).
Following this calculation, the BGE would vary between 1-21 % and 2 -13 % in the depth range of MLD-40 m and
40-60 m, respectively, being on average almost constant over the two different depth ranges (6.6 and 5.0%). ii)
Applying a BGE in the range of 10 and 30% and the measured bacterial production, the calculated bacterial DOC
$uptake_\phi$ was 0.08-7.10 mmol C $m^{-3}$ $d^{-1}$. Hence, respiration of DOC to $CO_2$ accounted for a $BOD_\phi$ of 0.06-6.39 mmol
$O_2$ $m^{-3}$ $d^{-1}$ (Table 1).

## 4. Discussion

We investigated bacterial degradation of DOM by measuring bacterial production as an estimate for organic carbon
transformation into biomass as well as rates of extracellular hydrolytic enzymes to provide information on the initial
steps of organic matter degradation (Hoppe et al., 2002). We expected reduced rates of organic matter degradation
within oxygen depleted waters, since reduced bacterial degradation activity might explain enhanced carbon fluxes in
suboxic and anoxic waters (Devol and Hartnett, 2001). However, although bacterial production decreased with depth
(Fig. 4b), this decrease was not related to oxygen concentrations. Moreover, no significant increase in bacterial
production was observed at the lower oxycline, when oxygen concentration increased again (Fig. 4b). Decreasing
bacterial production with depth has also been observed for fully oxygenated regions in the Atlantic (Baltar et al.,
2009) and the equatorial Pacific (Kirchman et al., 1995) and has been explained by a decrease in the amount of
bioavailable organic matter over depth.
The hypothesis of reduced bacterial degradation activity within the OMZ also implies reduced extracellular enzyme
rates for the hydrolysis of organic matter. The extracellular enzymes rates of our study have to be interpreted
carefully since incubation was not fully anoxic and the remaining oxygen might have biased the results. Still, we
assume that most extracellular enzymes were present at the time of sampling and thus oxygen contamination during
the incubations did not strongly influence the rate measurements. In our study, neither GLUCase nor LAPase $V_{max}$
were reduced within the suboxic waters compared to the oxyclines irrespective of incubation conditions (Fig. 5d, e,
supplementary Fig. 3 and 4). Thus, our findings show no evidence for reduced organic matter degradation in suboxic
waters and are in good agreement with studies, which report similar bacterial degradation rates for oxic and suboxic
waters (Cavan et al., 2017; Lee, 1992; Pantoja et al., 2009). Consequently, the hypothesis of enhanced carbon export
in OMZ waters due to reduced organic matter degradation seems fragile and alternative explanations for enhanced
carbon export efficiency e.g. reduced particle fragmentation due to zooplankton avoiding hypoxia (Cavan et al.,
2017) may be more likely. Likewise, a reduced degradation of particulate organic carbon in suboxic waters as it is
often assumed in global ocean biogeochemical models may have to be reconsidered (Ilyina et al., 2013).
Within OMZs dissolved nitrogen fuels e.g. denitrification or anaerobic ammonium oxidation (anammox) and is
reduced to e.g. dinitrogen gas that evades to the atmosphere. Current estimates result in 20-50% of the total oceanic
nitrogen loss occurring in OMZs (Lam and Kuypers, 2011). Meanwhile, a preferential degradation of amino acid
containing organic matter in suboxic waters compared to oxic waters has been suggested (Van Mooy et al., 2002).
Degradation of nitrogen compounds by heterotrophic bacteria (e.g. denitrifiers) in suboxic waters enables the release
of ammonia and nitrite and subsequently may support anammox, an autotrophic anaerobic pathway (Babbin et al.,
2014; Kalvelage et al., 2013; Lam and Kuypers, 2011; Ward, 2013). This interaction between denitrifiers and
anammox bacteria could fuel the loss of nitrogen to the atmosphere. Our data indeed showed enhanced degradation
of amino-acid-containing organic matter in low oxygen waters. Indicators for protein decomposition, i.e. LAPase
$V_{max}$ and the degradation rate of DHAA by LAPase, were more pronounced within the suboxic waters (Fig. 5b, d).
Therefore, observed LAPase rates are in line with the hypothesis of preferential degradation of nitrogen compounds
under suboxia. However, simultaneous rate measurements of protein hydrolysis, nitrate reduction (e.g.
denitrification) and anammox are needed to prove an indirect stimulation of anammox by protein hydrolysis via
denitrification. A close coupling between anammox and nitrate reducing bacteria has previously been shown for
wastewater treatments. There, nitrate reducers directly take up organic matter excreted by the anammox bacteria
which in turn benefit from the released nitrite by respiratory nitrate reduction (Lawson et al., 2017). In the Pacific,
denitrifiers and anammox bacteria are separated in space and time (Dalsgaard et al., 2012), potentially weakening a
direct inter-dependency.
To investigate physiological effects of suboxia, we normalized bacterial production and enzymatic rates to cell
abundance and found higher cell-specific bacterial production near the oxycline compared to suboxic waters and
highest cell-specific enzyme rates at the lower oxycline (Fig. 4d, 5g-j). Higher cell-specific bacterial production at
oxic-anoxic interfaces in the water column has previously been reported for the Baltic Sea (Brettar et al., 2012).
Baltar et al. (2009) showed increasing cell-specific enzymatic rates and decreasing cell-specific bacterial production,
with increasing depth in the subtropical Atlantic and related this pattern to decreasing organic matter lability. In our
study, differences in cell-specific bacterial production between suboxic waters and the oxycline did not persist at the
most coastal stations (G and T). This indicates the stimulation of bacterial activity, including anaerobic respiratory
processes, by the high input of labile organic matter. Therefore, our study suggests that a possible impairment of cell-
specific bacterial production under suboxia is reduced by supply of organic matter. However, this hypothesis is
restricted to a very limited number of samples and should be tested in further studies. While labile organic matter is
decreasing with depth (e.g. Loginova et al., 2019), TDN (Fig. 3c), especially inorganic nitrogen is increasing with
depth. Thus, high concentrations of inorganic nitrogen at the lower oxycline are available for heterotrophic and
chemoautotrophic energy gains. For instance, the co-occurrence of nitrate reduction, that was still detected at 25
$\mu$mol $O_2$ $L^{-1}$, and microaerobic respiration might have stimulated cell-specific production or the accumulation of
especially active bacterial species (Kalvelage et al., 2011, 2015).
Depth distribution of cell-specific and total bacterial production was different (Fig. 4b, d and supplementary Fig. 2);
cell-specific production was significantly reduced in suboxic waters, while total production was more similar in
suboxic waters compared to the oxycline. This suggests that lower cell-specific production was compensated by
higher cell abundance within the suboxic waters (Fig. 4c), resulting in an overall unhampered bacterial organic
matter cycling in the OMZ core. One reason for the accumulation of cells within the OMZ might be reduced
predation, suggesting the OMZ core as an ecological niche for slowly growing bacteria. Reduced grazing by
bacterivores thus preserves bacterial biomass in suboxic waters from entering into the food chain. This way of
bacterial biomass preservation has been suggested as possible explanation for enhanced carbon preservation in
anoxic sediments by Lee (1992), and may also explain our observations for the anoxic water column.
In general, bacterial community composition in OMZs has been shown to be strongly impacted by oxygen. In the
OMZ near the shelf off Chile Arctic96BD-19 and SUP05 dominate heterotrophic and autotrophic groups in hypoxic
waters (Aldunate et al., 2018). Next to the appearance of autotrophic bacteria that are related to sulphur (e.g. SUP05)
or nitrogen cycling (e.g. Planctomycetes), also bacteria related to cycling of complex carbohydrates have been
discovered in OMZs (Callbeck et al., 2018; Galán et al., 2009; Thrash et al., 2017), and may explain the unaltered
high potential ($V_{max}$) of the extracellular enzymes GLUCase and heterotrophic bacterial production in suboxic waters
in our study (Fig. 5e, 4b). For instance, SAR406, SAR202, ACD39 and PAUC34f have the genetic potential for the
turnover of complex carbohydrates and anaerobic respiratory processes, in the Gulf of Mexico (Thrash et al., 2017).
Consequently, our findings of active bacterial degradation of DOM are supported by molecular biological studies.
Still, simultaneous measurements of bacterial degradation and production have to be combined with molecular
analysis, in future studies off Peru.
Heterotrophic bacteria are the main users of marine DOM (Azam et al., 1983; Carlson and Hansell, 2015) and
responsible for ~79% of total respiration in the Pacific Ocean (Del Giorgio et al., 2011), proposing that heterotrophic
bacteria drive organic matter and oxygen cycling in the ocean and significantly contribute to the formation of the
OMZ. Under the assumption that the calculated loss of DOC during diapycnal transport (<60 m) is caused solely by
bacterial uptake and subtracting the amount of carbon channelled into biomass production, our study verifies the
importance of bacterial DOC degradation for the formation of the OMZ. We estimated a BOD (0.98-3.36 mmol $O_2$
$m^{-3}$ $d^{-1}$) that is in line with earlier respiration measurements in the upper oxycline off Peru (Kalvelage et al., 2015)
and represents 18-33% of the oxygen loss over depth, implying a rather low average BGE (6.5 and 5.0 %) (Table 1).
Calculating the bacterial uptake of DOC from production rates and a more conservative BGE between 10 and 30% as
previously suggested (Rivkin and Legendre, 2001) for the *in situ* temperature of 14 to 19 °C, 3-209% of the DOC
loss and 1-62% of oxygen loss could be attributed to bacterial degradation of DOM. The first approach reveals an
average BGE (6.5 and 5.0%) that is still within the range of previous reports for upwelling systems of the Atlantic
(<1-58%) and northeastern Pacific (<10%) (Alonso-Sáez et al., 2007; Del Giorgio et al., 2011). The high variability
in BGE is a topic of ongoing research. Until now 54% of the variability could be explained by variations in
temperature (Rivkin and Legendre, 2001). Our data suggest that oxygen availability may be another control of BGE
leading to rather low BGE in low oxygen waters. This is especially indicated by a low but rather constant average
BGE (6.5 and 5.0%), which we estimated for the water column down to 60 m depth under the assumption that all
DOC that is lost over depth can be attributed to bacterial uptake. A low BGE might be explained by a bacterial
community that has higher energetic demands, but in return is adapted to variable oxygen conditions. Additionally,
the BGE is decreasing with an increasing carbon to nitrogen ratio of the available substrate (Goldman et al., 1987).
In the OMZ off Peru the ratio between DOC and dissolved organic nitrogen is frequently high (~12 to 16) (Loginova
et al., 2019), and might further contribute to the low BGE. High respiration rates induced by bacterial DOC
degradation contribute to sustaining the OMZ, besides oxygen consumption by bacteria that hydrolyze and degrade
particulate organic matter (Cavan et al., 2017). Another, but likely minor contribution to overall respiration is made
by zooplankton and higher trophic levels (e.g. Kiko et al., 2016). Additionally, physical processes such as an
intrusion of oxygen depleted waters by eddies, upwelling or advection, may add to the oxygen and DOC loss over
depth (Brandt et al., 2015; Llanillo et al., 2018; Steinfeldt et al., 2015).
Uncertainties of our assumption that the loss of DOC is caused solely by bacterial uptake include other processes
potentially contributing to DOC removal, but not taken into consideration here like DOC adsorption onto particles,
DOC uptake by eukaryotic cells or the physical coagulation of DOC into particles, e.g. by formation of gel-like
particles such as transparent exopolymer particles and Coomassie stainable particles (Carlson and Hansell, 2015;
Engel et al., 2004, 2005). Moreover, temporal variations in diapycnal fluxes may be large, as indicated by the
confidence interval of solute fluxes (Fig. 2b, c) during this study and by 2 to 10 times lower DOC and oxygen loss
rates during other seasons (Loginova et al., 2019). However, our study is the first combining physical and microbial
rate measurements and gives estimates for carbon and oxygen losses in the upwelling system off Peru and can help
improving current biogeochemical models by constraining bacterial DOM degradation.
Loginova et al. (2019) conducted similar physical rate measurements in the same study area with ~2 and ~10 times
lower DOC and oxygen loss in the upper ~40 m compared to our study. Differences in loss rates were mainly caused
by a ~ 10 times higher diapycnal diffusivity of mass in our study. This may have been caused by weaker
stratification in the upper 100 m depth or differences in the turbulence conditions. Loginova et al. (2019) estimated a
contribution of bacterial DOM degradation to oxygen loss (38 %) based on the loss of labile DOC (DHAA and
DCHO). This value agrees well with our estimates of 18-33% of total oxygen loss, calculated under the assumption
that DOC loss is solely attributed to bacterial degradation. However, the comparison of DOC and oxygen loss within
each study revealed different patterns. Loginova et al. (2019) found a loss of DOC that clearly exceeded the loss of
oxygen within the upper ~40 m. Hence, respiration of DOC could fully explain the observed oxygen loss in that
study. In our study, more oxygen than DOC was lost over depth (Table 1). This loss of oxygen needs additional
explanations such as degradation of particulate organic matter and physical mixing processes. One reason for the
observed differences between the two studies that have been conducted in the same region might be seasonality. The
study by Loginova et al. (2019) took place in austral summer, whereas our data were gained during austral winter.
Water temperature was quite similar during both studies, probably due to the coastal El Niño one month before our
sampling campaign (Garreaud, 2018). Still, the study by Loginova et al. (2019) included more stations with high Chl
$a$ concentrations (~8 µg L$^{-1}$), as typical for the austral summer, indicating a more productive system with more labile
DOM (DCHO and DHAA). Prevalence of more labile DOM might explain the higher contribution of microbial
DOM respiration to oxygen loss in the study by Loginova et al. (2019).

In oxygen depleted waters of the Peruvian upwelling system, the chemoautotrophic process of anammox has been
assumed to dominate anaerobic nitrogen cycling (Kalvelage et al., 2013), with lower but more constant rates
compared to more sporadically occurring heterotrophic denitrification (Dalsgaard et al., 2012). Studies based on the
stoichiometry of organic matter suggest a general dominance of denitrification in relation to anammox and relate
variable ratios between these two processes to the stoichiometry of locally available organic matter (Babbin et al.,
2014; Ward, 2013). Our study points towards a widespread occurrence of heterotrophic anaerobic processes such as
denitrification or sulfate reduction (Canfield et al., 2010) in the Peruvian OMZ, since the here applied method for
measuring bacterial production is restricted to heterotrophs. Our rates for bacterial production within the suboxic
waters averaged to 37 $\mu$mol C m$^{-3}$ d$^{-1}$ (0.3-281 $\mu$mol C m$^{-3}$ d$^{-1}$).

We compared bacterial production, i.e. rates of carbon incorporation, with denitrification rates previously reported
for the South Pacific. Therefore, we converted one mol of reduced nitrogen that were measured by Dalsgaard et al.
(2012) and Kalvelage et al. (2013) to 1.25 mol of oxidized carbon after the reaction equation given by Lam and
Kuypers (2011). This calculation indicates that on average $\leq$19 $\mu$mol C m$^{-3}$ d$^{-1}$ are oxidized by denitrifying bacteria
in the Eastern Tropical Pacific (Dalsgaard et al., 2012; Kalvelage et al., 2013).
The amount of carbon oxidized by denitrification based on the studies of Dalsgaard et al. (2012) and  Kalvelage et al.
(2013) can be converted into bacterial production applying a BGE. The average temperature dependent BGE was
20%. A BGE of 20% agrees well with other studies (Del Giorgio and Cole, 1998). Assuming a BGE of 20%, the
denitrification rates of Dalsgaard et al. (2012) and  Kalvelage et al. (2013) suggest a bacterial production of $\leq$5 $\mu$mol
C m$^{-3}$ d$^{-1}$, equivalent to only about 14% of total average heterotrophic bacterial production in suboxic waters
determined in our study. For the sum of anaerobic carbon oxidation rates including denitrification, DNRA and
simple nitrate reduction, 109 $\mu$mol C m$^{-3}$ d$^{-1}$ (6-515 $\mu$mol C m$^{-3}$ d$^{-1}$) may be expected for the Peruvian shelf, with the
reduction of nitrate to nitrite representing the largest proportion (2-505 $\mu$mol C$^{-1}$ m$^{-3}$ d$^{-1}$), based on the relative
abundance of the different N-functional genes (Kalvelage et al., 2013). These anaerobic respiration measurements
are equivalent to a bacterial production of $\sim$ 27 $\mu$mol C m$^{-3}$ d$^{-1}$ (1-129 $\mu$mol C m$^{-3}$ d$^{-1}$) and are thus lower than our
direct measurements of bacterial production rates. Moreover, the reduction of nitrate, could not be detected at every
depth and incubation experiments partly showed huge variations over depth (Kalvelage et al., 2013), whereas we
were able to measure bacterial production in every sample. The same calculation can be repeated assuming a BGE of
6%, which is the average BGE within this study based on DOC loss and bacterial production. Assuming a BGE of
6%, the estimated 109 $\mu$mol C m$^{-3}$ d$^{-1}$ that are respired by anaerobic carbon oxidation (Kalvelage et al., 2013) would
represent 94% of the carbon uptake. Consequently, 7 $\mu$mol C m$^{-3}$ d$^{-1}$, i.e. 6% of the carbon uptake, are incorporated
into the bacterial biomass. A bacterial biomass production of 7 $\mu$mol C m$^{-3}$ d$^{-1}$ is even lower than the bacterial
production of 27 $\mu$mol C m$^{-3}$ d$^{-1}$, based on a BGE of 20% and cannot explain the average bacterial production
measured in suboxic waters during our study (37 $\mu$mol C m$^{-3}$ d$^{-1}$). Therefore, this estimation suggests higher rates of
heterotrophic anaerobic respiratory processes than previously measured. Since denitrification rates were not
measured directly, the comparability of published denitrification rates and our measurements of bacterial production
are limited. However, our data suggest that the carbon oxidation potential off Peru is more evenly horizontally and
vertically distributed than expected and also corroborate earlier suggestions of unexpectedly high rates of
heterotrophic nitrogen cycling in the OMZ off Peru based on observations of high concentrations of atmospheric
nitrous oxide (Bourbonnais et al., 2017).
**5.  Conclusion**
Our study suggests that suboxia does not reduce bacterial degradation of organic matter in the Eastern Tropical South
Pacific off Peru. Bacterial species are seemingly adapted to these environments and higher cell abundance

compensates for hampered cell-specific bacterial production under suboxia. Therefore, the previously observed enhanced carbon export in OMZs compared to oxygenated waters requires alternative explanations. Differences between cell-specific and total rates of bacterial activity allude to different controls of cell abundance in suboxic systems, highlighting the OMZ as a specific ecological niche. The combination of bacterial and physical rate measurements suggests that low BGEs in the upper oxycline contribute to sustaining the OMZ. Meanwhile, new findings during our study call for additional studies: i) DOC loss differed strongly between our investigation and the study of Loginova et al. (2019). Therefore, combined physical and biological rate measurements in the Peruvian upwelling system should be repeated during austral summer, to learn more about the interplay of DOC loss and bacterial production during different seasons. ii) Integrated measurements of denitrification, microaerobic respiration and bacterial production are needed to estimate the fractions of incorporated and respired carbon under suboxia. The BGE received in that way could support or disprove the low BGE estimate, which was calculated from DOC loss and bacterial production in our study. Consequently, our study highlights the need for a better mechanistic understanding and quantification of processes responsible for oxygen and DOM loss in OMZs that is inevitable to predict future patterns of deoxygenation in a warming climate.

*Data Availability.* PANGEA: 10.1594/PANGAEA.891247

*Author contributions.* M.M. and A.E. designed the scientific study, analysed the data and wrote the manuscript. J.L. calculated DOC and oxygen fluxes, G.K. sampled and calibrated the CTD data and both J.L. and G.K. commented on the manuscript.

*Competing interests.* The authors declare that they have no conflict of interest.

*Acknowledgments:* We thank Jon Roa, Tania Klüver and Ruth Flerus for the sampling and/or analysis of DOC/TDN; cell abundance, bacterial production and DHAA. Moreover, we would like to thank Judith Piontek, Sören Thomsen, Carolina Cisternas-Novoa and Frédéric A.C. Le Moigne who helped and gave advice for sampling during the cruises. We are grateful to the working group of Hermann Bange and Stefan Sommer who provided Winkler measurements. We thank the cruise leaders Hermann Bange and Marcus Dengler, crew, officers and the captains of the F.S. Meteor for the support on board and the successful cruises. This study was supported by the Helmholtz Association and by the Collaborative Research Center 754 / SFB Sonderforschungsbereich 754 'Climate-Biogeochemistry Interactions in the Tropical Ocean'.

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

**Figure legends**

**Figure 1:** Station map. All presented stations in the Eastern Tropical South Pacific off Peru sampled in 2017. For detailed
informations about the stations see supplementary Table 1.
**Figure 2**: Measured concentrations and calculated proxies for the change of dissolved organic carbon (DOC) and dissolved
oxygen (DO) flux over depth for stations G-T: The average diapycnal diffusivity of mass ($K_\rho$) over depth with confidence interval
and the constant $K_\rho$ ($1 \times 10^{-3}\ m^2 s^{-1}$ ) that was used for further calculations **(a)**. Concentrations of DOC in the upper 100 m and
the resulting change of DOC flux over depth ($\nabla \Phi$) **(b)**. Concentrations of DO in the upper 100 m and the resulting change of DO
flux over depth ($\nabla \Phi$) **(c)**.
**Figure 3**: Biotic and abiotic conditions at selected stations exemplary for the sampling conditions. Chlorophyll **(a)**, temperature
**(b)**, total dissolved nitrogen (TDN) **(c)**, dissolved organic carbon (DOC) **(d),** carbon content of dissolved hydrolysable amino
acids (DHAA) **(e)** and carbon content of high molecular weight dissolved carbohydrates (DCHO) **(f)** over depth at different
stations from on- to offshore off Peru.
**Figure 4:** Bacterial growth activity at different *in situ* oxygen concentrations from on- to offshore off Peru during April 2017
(M136). Oxygen concentrations **(a)**, total bacterial production (BP) **(b)**, bacterial abundance **(c)** cell-specific BP **(d)** over the
upper 800 m depth with a zoom in the upper 100 m (small plots).
**Figure 5:** Extracellular enzyme rates at different *in situ* oxygen concentrations during April and June 2017 (M136, M138).
Oxygen concentrations **(a)**, degradation rates of dissolved amino acids (DHAA) by leucine-aminopeptidase (LAPase) **(b)**,
degradation rates of high molecular weight dissolved carbohydrates (DCHO) by ß-glucosidase (GLUCase) **(c)** total potential
LAPase rates ($V_{max}$) **(d)**, Glucase $V_{max}$ **(e)**, cell abundance **(f)**, cell-specific degradation rates DHAA by LAPase **(g)**, cell-specific
degradation rates of DCHO by GLUCase **(h)**, cell-specific LAPase $V_{max}$ **(i)** and cell-specific Glucase $V_{max}$ **(j)** at different oxygen
regimes off Peru.



**Tables**
**Table 1:** Estimates of oxygen and DOC loss over depth based on *in situ* physical observations and bacterial rate measurements. Oxygen and DOC loss rates (mmol m$^{-3}$ d$^{-1}$) were
estimated from the change in oxygen and DOC fluxes over depth. The bacterial uptake of DOC (mmol m$^{-3}$ d$^{-1}$) was calculated from bacterial production (mmol m$^{-3}$ d$^{-1}$) based on a
growth efficiency of 10 and 30% (DOC uptake$_\phi$). The bacterial oxygen demand (BOD, mmol m$^{-3}$ d$^{-1}$) and bacterial growth efficiency (BGE$_\varepsilon$, %) was calculated from bacterial
production and the assumption that DOC loss can be completely explained by bacterial uptake (BOD$_\varepsilon$) or estimated based on a BGE of 10 and 30% (BOD$_\phi$).

| Depth | oxygen loss | DOC loss | DOCuptake$_{\phi10}$ | | | DOC uptake$_{\phi30}$ | | | Bacterial Production | | | BOD$_\varepsilon$ | | | BOD$_{\phi10}$ | | | BOD$_{\phi30}$ | | | BGE$_\varepsilon$ | | |
|---|---|---|---|---|---|---|---|---|---|---|---|---|---|---|---|---|---|---|---|---|---|---|---|
| | avg | avg | avg | min | max | avg | min | max | avg | min | max | avg | min | max | avg | min | max | avg | min | max | avg | min | max |
| MLD-40 | 10.23 | 3.4 | 2.22 | 0.35 | 7.10 | 0.74 | 0.12 | 2.37 | 0.22 | 0.03 | 0.71 | 3.17 | 2.68 | 3.36 | 2.00 | 0.31 | 6.39 | 0.52 | 0.08 | 1.66 | 6.55 | 1.02 | 20.92 |
| 40-60 | 5.55 | 1.13 | 0.56 | 0.25 | 1.46 | 0.19 | 0.08 | 0.49 | 0.06 | 0.03 | 0.15 | 1.07 | 0.98 | 1.10 | 0.51 | 0.23 | 1.32 | 0.13 | 0.06 | 0.34 | 5.00 | 2.26 | 12.97 |




**Figures**

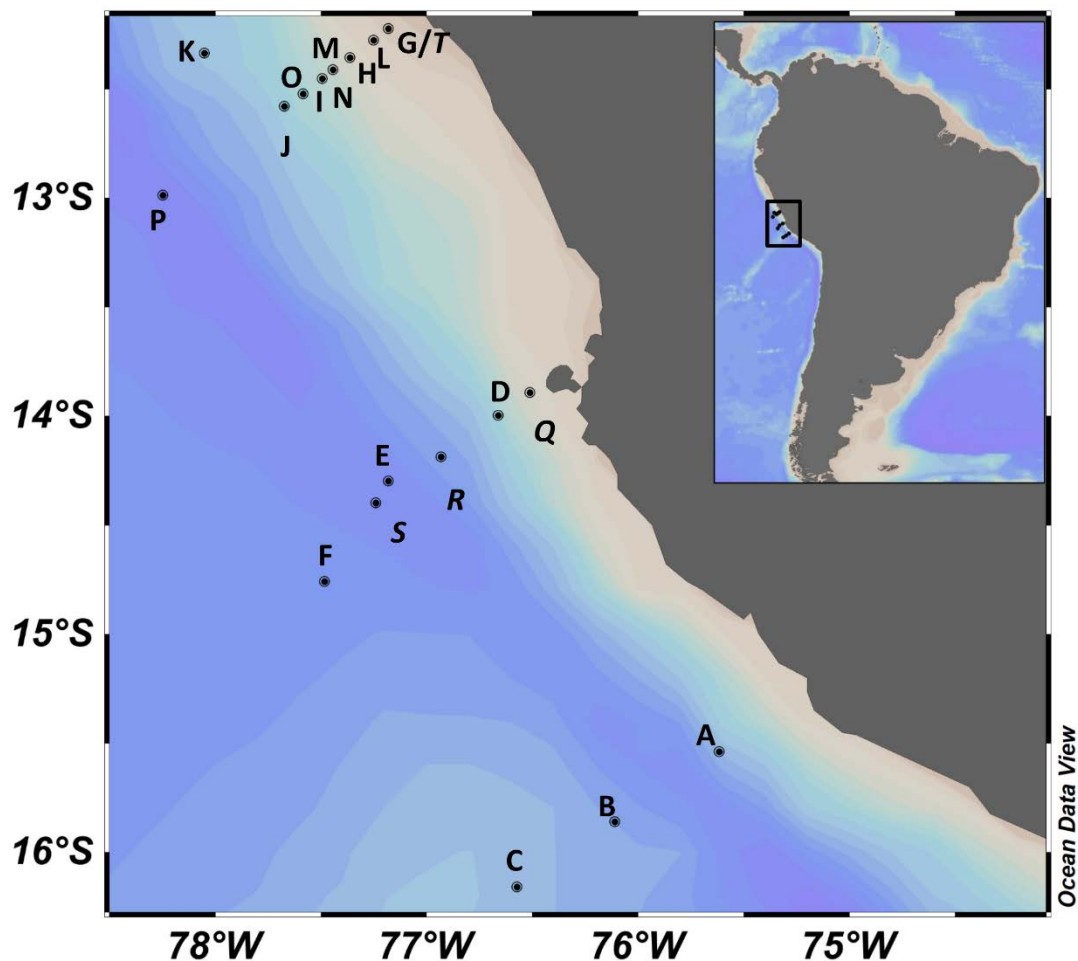


**Figure 1**

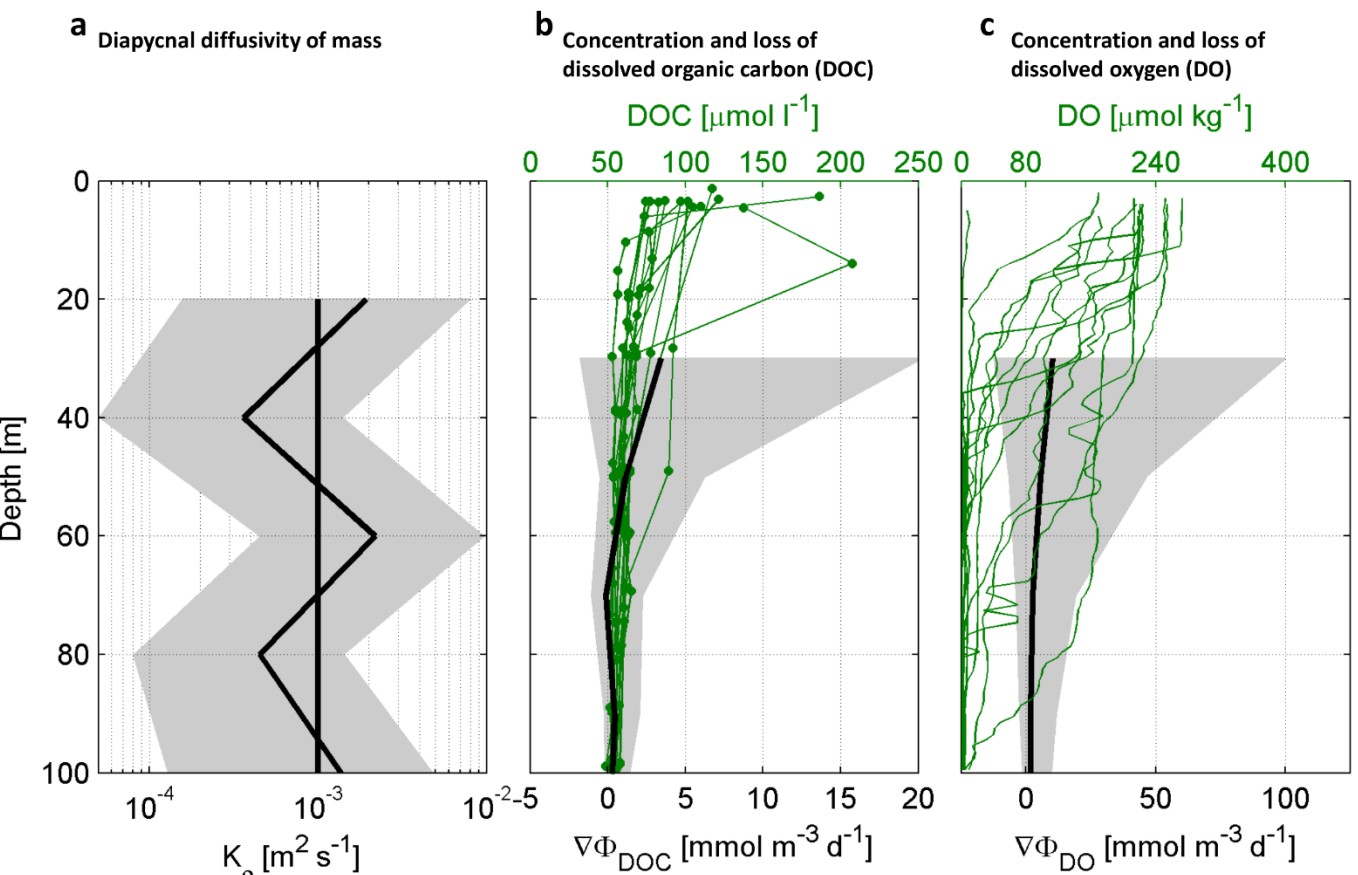


**Figure 2**

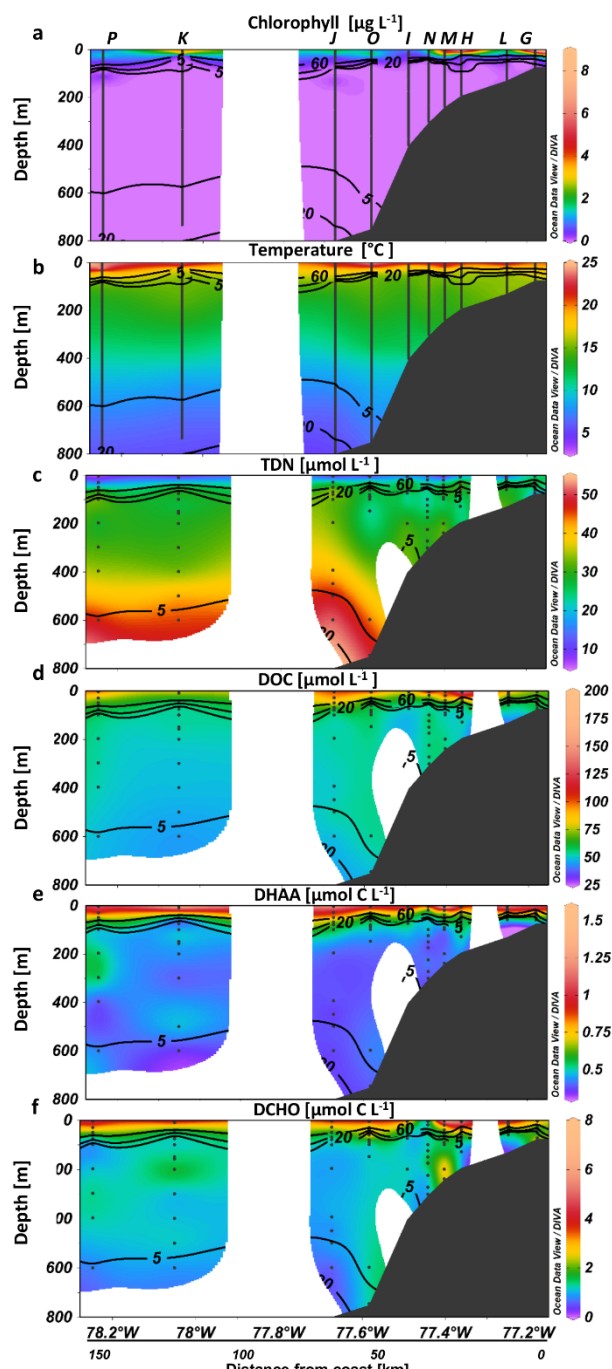


**Figure 3**

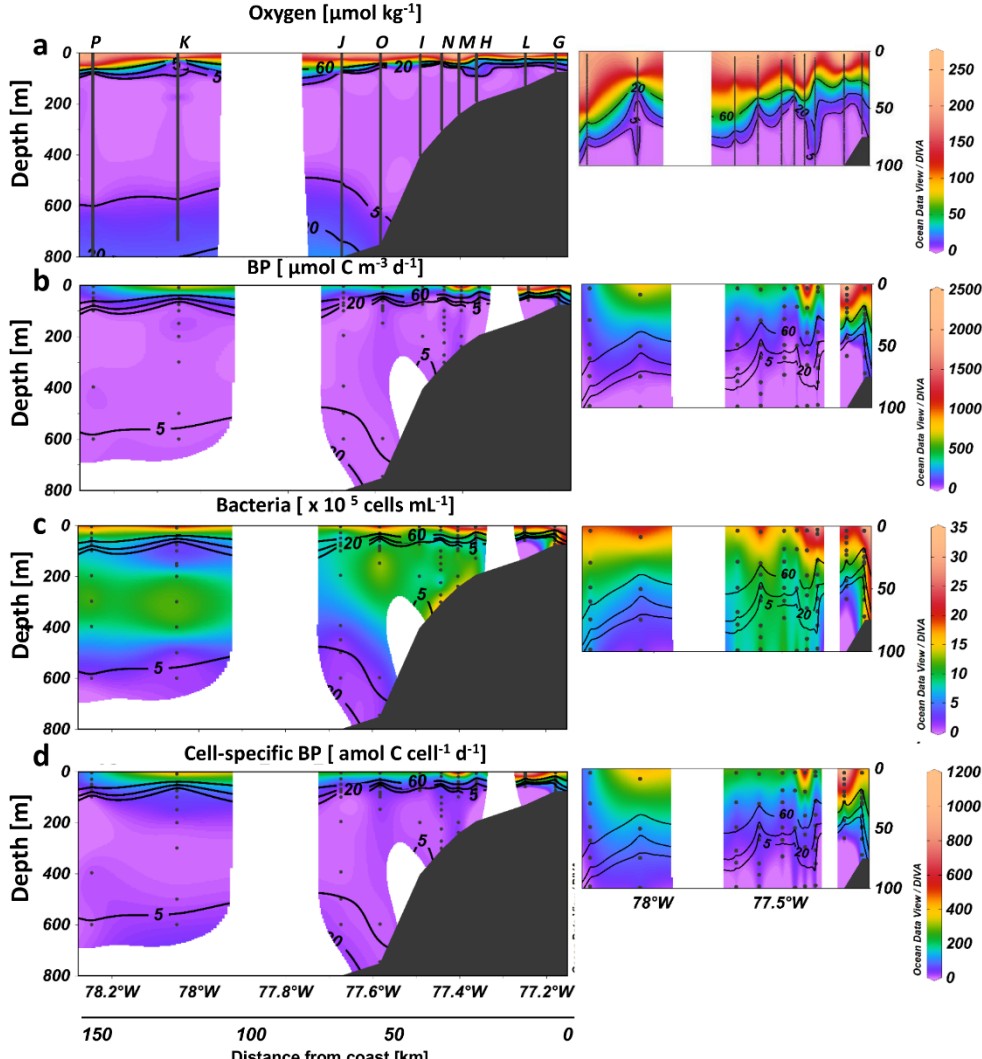


**Figure 4**

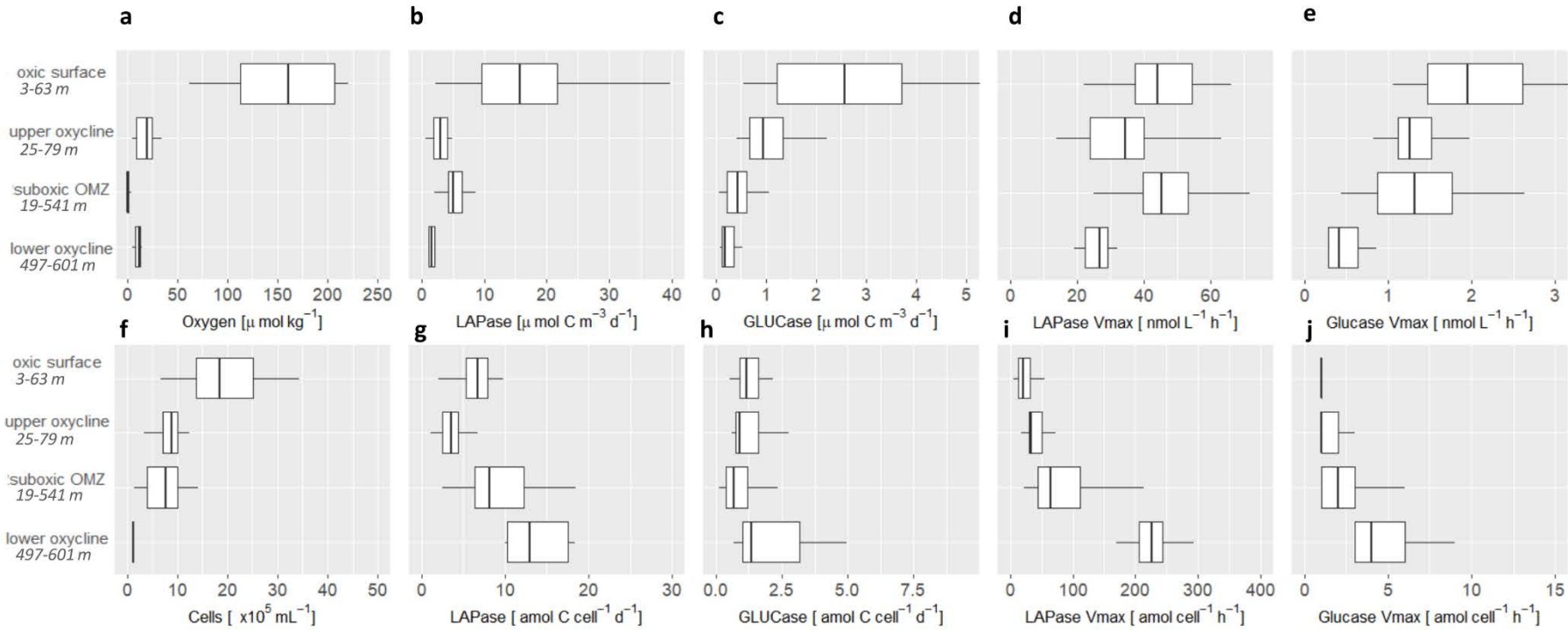


**Figure 5**


