# Peer review of "Bacterial degradation activity in the Eastern Tropical South"

_Biogeosciences, 2019_

## Short Comment (SC1) · 17 Jun 2019

The statement on lines 54-56 is a bit misleading. While it's true that Taylor et al. (2009) report higher volumetric peptidase rates in oxic waters than in anoxic waters of the Cariaco Basin, it appears this is attributable to higher particulate organic matter concentrations in oxic waters. If you refer to Figure 9 in cited paper, you will see that when peptidase activity is normalized to particulate nitrogen, it becomes indistinguishable in oxic and anoxic samples. The same is true for the other three ectohydrolases studied in the Taylor et al. (2009) paper. I think this is key to the points you raise in your current manuscript and warrants clarification.

---

## Short Comment (SC2) · 18 Jun 2019

Might I suggest? "Hydrolytic rates in the Cariaco Basin were significantly higher in oxic compared to anoxic waters, however this difference did not persist after rates were normalized to particulate organic matter concentrations (Taylor et al., 2009)."

---

## Author Comment (AC1) · 18 Jun 2019

Dear Gordon Taylor,

Thank you for pointing to this misleading sentence, since the term "volumetric rate" does not explicitly express that the differences between hydrolytic rates in oxic and anoxic waters was explained by different particulate organic matter concentrations in the reference Taylor et al. 2009. Therefore, we will add more detailed information in the following version of our manuscript:

"For instance, volumetric rates of protein hydrolysing enzymes (peptidases) did not

differ between oxic and anoxic incubations of Baltic Sea water (Hoppe et al., 1990). Hydrolytic rates in Cariaco waters were significantly higher in oxic compared to anoxic waters, however this difference did not persist after the normalization to particulate organic matter concentrations (Taylor et al., 2009)."

---

## Referee Comment (RC1) · Anonymous Referee #1 · 24 Jul 2019

General Comments.

The manuscript by Maßmig et al. shows interesting results from two cruises in the ETSP OMZ off Peru. The combination of DOC, TDN, DHAA and DCHO with bacterial production and extracellular enzyme rates provides a nice overview of the microbial activity in general terms. Authors also show diapycnal fluxes for oxygen and DOC, including the potential role of microbial processes into those total fluxes. A similar manuscript has been recently published by the same authors (Loginova et al. 2019 Biogeosciences, 16). DOC, TDN, DON, DHAA, DCHO and diapycnal DOC and oxygen fluxes were also measured/estimated in a previous cruise in the same area. It is clear

that the present study includes other data but discussion lacks a comparison between both studies and some results/conclusions seems to be repeated. For instance, the 33% of oxygen loss over depth attributed to bacterial oxygen demand is quite similar than in the previous study (38%). Please extend the discussion and comparison with the previous manuscript.

The stations were sampled in two cruises (April and June) and distributed in three transects perpendicular to the coast: Lima, Paracas and Puerto Caballas (approx.). Spatial and temporal variability is however not considered in the manuscript. Some data correspond to some transects and cruise and other data correspond to other but no clear differentiation is included. Substances concentrations and fluxes were measured in Lima and Paracas transects in April, but enzymatic activity was measured in Paracas and Puerto Caballas in June. These data are however pooled and used for all the later estimations without any further consideration of spatiotemporal differences. Only one transect (Lima) is shown in Figures 2-3, are the conditions equal in the other transects (Temperature, Oxygen, Chlorophyl. . .)?

Specific Comments.

Title: It does not reflect the measurements performed in the study. "Bacterial organic carbon uptake" was not measured.

L19: Bacterial growth efficiency was taken from Rivkin and Legrende (2001) as a simple function of temperature. It should not be considered as a result from the present study.

L25: Gruber et al. is a good reference for global scale processes and future conditions, however, a better reference for the measurement of anoxic conditions in the ETNP OMZs would be: Tiano et al. 2014. Deep-Sea Res. Part I. 94, 173-183.

L28: One classical reference dealing with the extention and volumens of the different OMZs is Paulmier & Ruiz-Pino 2009. Progress in Oceanography 80, 113-128.

[Figure]

L36-37: DNRA might result in lower metabolic energy yield, but it is not a mayor pathway in OMZs. Although it has been found in the ETSP, it showed sporadic and low rates (Kalvelage et al. 2013). On the other hand, denitrification might be considered one of the main anaerobic heterotrophic process but it is yielding 99% of the energy compared to aerobic respiration, i.e. it is almost equally efficient. This paragraph seems to be biased to introduce the idea of inefficient anaerobic metabolism, but it is not proved.

L51-58: The effect of oxygen concentration on bacterial production and extracellular enzymes activity was ambiguous before the comment of G.Taylor. When the differential particulate organic matter was considered, hydrolytic rates were similar. This paragraph needs then some rewording because the study is not clearly justified now.

L61-62: Again, I disagree with the "lower efficiency of anaerobic respiration" (unless other processes different than denitrification are proved to be relevant).

L86-87: It is not clear for me if the filter or the ampule were rinsed with the sample.

L96-106: To be consistent, what is the detection limit and precision of the DHAA and DCHO analysis?

L116: Fig. 5 is cited before Fig. 2.

L131: Bacterial Production was measured at 13°C for all samples. Considering the range of temperatures found along the water column (7-24 °C), incubation temperature was up to 12°C off the in situ temperature. There were no compensation for the temperature variation, probably leading to significant deviation from in situ estimates. Considering the relevance of these results for the discussion, authors should correct measured rates with in situ temperature.

L154: Enzymatic rates were also measured at a fixed temperature of 13°C. Could in situ temperature be taken into account?

L159-160: Please improve the description of the "Gas tight incubator". Considering the oxygen concentration values in your "low oxygen" incubations (8-40 umol/kg), how

realistic are the conclusions applied to the anoxic core from these incubations? Oxygen concentrations of 8 uM are way above the Km for microbial processes such as Oxygen respiration, ammonium and nitrite oxidation, for instance, and above the inhibition values for anammox and denitrification. Please, include in the discussion the possible limitation of the measurements considering the high oxygen values achieved in the incubations.

L201 (and L314): TDN includes the inorganic fraction. Nitrate in OMZs increases with depth, and might reach values up to 30-40 uM (example: Lam et al. 2009. Proceedings of the National Academy of Sciences 106, 4752-4757), which might represent 80-100% of the measured TDN. Could the authors include inorganic nutrients and use DON instead?

L261-270: It is not clear how the parameters (DOC loss) have been calculated, only ranges are shown and it feels like the ranges have been subtracted without including the apparent heterogeneity of the different stations. Based on the data shown in Fig. 5, the large differences in the oxyclines must result in large differences in diapycnal oxygen fluxes. Some separation in the data shown in Fig. 5 would be advisable. Anoxic conditions are reached at depths varying from 20 to 100 m, probably with very different values for the measured variables (Chl a, DOC...) too. Contrary, DOC values change quickly in the first 10 m, but seems to be relatively constant below. Dots are not connected with lines so it seems to be a pool of data without a clear pattern. All the station seems to be the same.

L275: DNRA might have lower energy yield, it is not so low for denitrification.

L291-292: I would delete "nitrous oxide" otherwise further explanation is needed as the contribution from anammox to N2O production is quite reduced.

L296-297: Remove "respiratory" from "autotrophic anaerobic respiratory pathway". Babbin et al (2014, Science 344, 406-408) and Kalvelage et al. (2013. Nature Geosciences 6, 228-234) are also appropriate references for that quote. In addition, I would

delete the sentence in L298-299, denitrification+anammox are included in the global estimations for N losses.

L301-307: This section exceed the results obtained in the present manuscript. A possible link to N cycle could be pointed, but the connection between hydrolysis and coupled denitrification-anammox is not supported.

L314-316: Inorganic nitrogen might be the mayor fraction of TDN. This fact must be taken into account, especially if any stimulation of metabolism is considered.

L317-322 and L323-331: These paragraphs seem to be not finished. There are no clear conclusion for the discussion of these results.

L346-347: According to M&M, BGE followed the established temperature dependence. If no other parameter was used for its calculation, I cannot see how the results of this manuscript for this calculated (but not measured) parameter suggest that oxygen availability control bacterial growth efficiency.

L365-367: Well, this study provides estimations, but does not provide measurements for carbon and oxygen losses.

L378: Why a BGE of 20% is now assumed? BGE was estimated based on in situ temperature before.

L383-390: The presented data for bacterial production can not be directly attributed to denitrification as it was not directly measured and the high oxygen levels during the BP measurements could have inhibited denitrification. The last and conclusive sentence seems to be pretentious.

L392-400: Conclusions should be more attached to the obtained and proved results of the measurements. The measurements of bacterial production do not allow to prove the dominance of individual pathways and even less to link it with the production of nitrous oxide.

---

## Referee Comment (RC2) · Anonymous Referee #2 · 20 Aug 2019

This is a useful study investigating the complicated microbial dynamics within oxygen minimum zones with many different biogeochemical and physical measurements made. The authors focus on calculating bacterial production predominantly associated with carbon cycling, but then also use other studies to consider the input of nitrogen cycling and anoxic processes. The manuscript is very well written, generally clear and detailed. I have a few suggestions to revise the text and figures to make some of the points clearer and to hopefully clarify some uncertainties. One point that was not mentionned was that the bacteria were collected from suboxic concentrations but rates measured in oxic conditions I assume? How might the fact the microbes are being oxidised affect your results? It is difficult to work in OMZs and many of the studies cited

would have done a similar thing but i think this should be discussed. Also the authors seemed to switch between top/bottom hypoxic and upper/lower oxycline, which i took to mean the same thing. If not this should be clarified.

Specific points: Line 16 - Change to 'from the upper AND lower oxyclines', using 'or' makes it seem negative and I had to read it a few times to understand you were saying production was high.

Line 73 - I noticed the transects had data from both cruises. They are quite close together temporally, but even so some discussion on how the data is aggregated and if temporal affects are accounted for is needed. Did you look at the data separately per cruise too? Which data/transects are used in the figures? What is the seasonality like in the region?

Line 139 - citation for first use of BOD and write in full first time used in main text

Line 145 - what temperature dependance? Is a conversion used? Cite

Line 198 - Perhaps place oxygen concentration in Fig. 2 with the other 'standard' oceanographic measurements, as it is a key plot for this paper. I would also add on horizontal layers onto the transect images, i.e. by using black lines to show oxy-cline/omz/hypoxic/oxic layers.

Also, what was lowest oxygen concentration, was it 5 umol/L so only suboxic or even lower to maybe anoxic as set by your definitions in the introduction? The study refers to 'suboxic' throughout which makes me think outside of the OMZ, but actually this is the OMZ. Being clear about this early on in the results would help.

Line 199 - Does this mean OMZ is 100-500 m depth, be explicit

Line 206 - 'except for most coastal stations' - what happened at these stations?

Line 235 - full statistical results in parentheses is great to see and the correct way to present results, however with so many tests and parts of the text in parentheses it stops

the flow when reading. Can you shorten the statistical results in some way? Or add a table to the supplementary material?

Line 242 - normalisation completely changes the pattern of production with depth and oxygen, reverses it compared to un-normalised data. It would be good to show this and discuss further, perhaps using scatter plots too.

Line 243 - Units of 'amol per cell per day' are incredibly low as one may expect from a 'per cell' measurement, but is this comparable with results from other studies?

Line 284 - Is this finding because experiments were run in oxic conditions, as were some of the studies you cited too. But should consider the affects of exposing microbes from OMZ to oxygen.

Line 375 - where did the data of amount of reduced nitrogen come from, was it Kalvelage? I found this section, whilst interesting, a little hard to follow which numbers were from this study and which from others. For instance, why use BGE from Del Giorgio 1998 when you calculated it in this study?

Line 388 - Do you mean distributed evenly vertically or horizontally, or both?

Line 399 - I agree with final sentence of paper but not mentioned anywhere how can improve understanding or quantification processes, so on its own this final sentence is a bit weak for such a thorough study.

Figures: Fig. 2 and Fig.3 - show horizontal oxygen regions as suggested above. Also, reduce extrapolation with ODV, large gap ∼ 100 km where no station/data between coastal and offshelf. Which interpolation did you use in ODV? The stations are running from the coast which is more east than offshore according to figure 1, so perhaps flip horizontally to reflect the east to west/coast to offshore nature of the spatial distribution. Having longitude instead of distance from coast (or both preferably) may be more useful, and make clear the inset is top 100 m.

Fig 4 - labels of oxygen regimes different to text where instead oxyclines often referred

to, is that the same as top and bottom hypoxic? Continuity throughout would be helpful.

Fig. 5. Add a title for each panel so do not need to refer to legend as much.

[Figure]

---

## Author Comment (AC2) · 13 Sep 2019

Anonymous Referee #1 (AR1): The manuscript by Maßmig et al. shows interesting results from two cruises in the ETSP OMZ off Peru. The combination of DOC, TDN, DHAA and DCHO with bacterial production and extracellular enzyme rates provides a nice overview of the microbial activity in general terms. Authors also show diapycnal fluxes for oxygen and DOC, including the potential role of microbial processes into those total fluxes. A similar manuscript has been recently published by the same authors (Loginova et al. 2019 Biogeosciences, 16). DOC, TDN, DON, DHAA, DCHO and diapycnal DOC and oxygen fluxes were also measured/estimated in a previous cruise

in the same area. It is clear that the present study includes other data but discussion lacks a comparison between both studies and some results/conclusions seems to be repeated. For instance, the 33% of oxygen loss over depth attributed to bacterial oxygen demand is quite similar than in the previous study (38%). Please extend the discussion and comparison with the previous manuscript.

Author Comment (AC): We thank the reviewer for this comment and will include the following paragraphs in the revised version of the manuscript. Additionally, we will include a paragraph concerning the seasonality of the Peruvian system in the introduction (see comment of second Reviewer concerning line 73):

"Loginova et al. (2019) conducted similar physical rate measurements in the same study area with ∼2 and ∼10 times lower DOC and oxygen loss in the upper ∼40 m compared to our study. Differences in loss rates were mainly caused by a ∼ 10 times higher diapycnal diffusivity of mass in our study. This may have been caused by weaker stratification in the upper 100 m depth or differences in the turbulence conditions. Loginova et al. (2019) estimated a contribution of bacterial DOM degradation to oxygen loss (38 %) based on the loss of labile DOC (DHAA and DCHO). This value agrees well with our estimates of 18-33% of total oxygen loss, calculated under the assumption that DOC loss is solely attributed to bacterial degradation. However, the comparison of DOC and oxygen loss within each study revealed different patterns. Loginova et al. (2019) found a loss of DOC that clearly exceeded the loss of oxygen within the upper ∼40 m. Hence, respiration of DOC could fully explain the observed oxygen loss in that study. In our study, more oxygen than DOC was lost over depth (Table 1). This loss of oxygen needs additional explanations such as degradation of particulate organic matter and physical mixing processes. One reason for the observed differences between the two studies that have been conducted in the same region, might be seasonality. The study by Loginova et al. (2019) took place in austral summer, whereas our data were gained during austral winter. Water temperature was quite similar during both studies, probably due to the coastal El Niño one month before our sampling campaign

(Gerreaud 2018). Still, the study by Loginova et al. (2019) included more stations with high Chl a concentrations (∼8 $\mu$g L-1), as typical for the austral summer, indicating a more productive system with more labile DOM (DCHO and DHAA). Prevalence of more labile DOM might explain the higher contribution of microbial DOM respiration to oxygen loss in the study by Loginova et al. (2019). Additionally, Loginova et al. (2019) sampled with a much higher vertical resolution within the upper 140 m, restricting the direct comparability with our study.

AR1: The stations were sampled in two cruises (April and June) and distributed in three transects perpendicular to the coast: Lima, Paracas and Puerto Caballas (approx.). Spatial and temporal variability is however not considered in the manuscript. Some data correspond to some transects and cruise and other data correspond to other but no clear differentiation is included. Substances concentrations and fluxes were measured in Lima and Paracas transects in April, but enzymatic activity was measured in Paracas and Puerto Caballas in June. These data are however pooled and used for all the later estimations without any further consideration of spatiotemporal differences. Only one transect (Lima) is shown in Figures 2-3, are the conditions equal in the other transects (Temperature, Oxygen, Chlorophyl: : :)?

AC: With our approach we focus on possible differences between oxygen regimes. Hence, statistics of bacterial production and of extracellular enzyme rates are always related to the different oxygen concentrations. However, we will include figures of oxygen and Chl a concentrations and temperatures for the remaining stations in the supplement. Moreover, a more differentiated description of the study site and a comparison between cruises will be included in section 3.1:

"During our two cruises to the Peruvian upwelling system (Fig. 1), seasonal variability caused higher maximum Chl a concentrations and warmer temperatures in April compared to June 2017. Chl a concentration reached up to 11 and 4 $\mu$g l-1 within the upper 25 m in April and June, respectively. Still, average Chl a concentrations within the upper 10 m (M136: 3.1±2.6 $\mu$g l-1; M138: 2.8 ± 1.3 $\mu$g l-1) were not significantly different

between the two cruises (nM136=75, n M138= 40, W=1416, p= 0.6). At depths >50 m, Chl a concentration was generally below detection limit (Fig. 2a, supplementary Figure 1). Within the upper 10 m the water was warmer in April (21.3 ±1.6°C) compared to June (17.6 ± 0.6°C) (nM136=75, n M138= 40, W=2886, p<0.01) (Fig. 2b, supplementary Figure 1). Oxygen >100 $\mu$mol kg-1 was observed in the surface mixed layer, decreased steeply with depth and reached suboxic concentrations (<5$\mu$mol L-1) at > 60 ± 24 m (Fig. 3a and 4a, supplementary Figure 1). Shallowest depth with suboxic oxygen concentrations were 14 m in April (station Q) and 29 m in June (station D), probably influenced by the distance from shore (Q<D). Oxygen increased again to up to 15 $\mu$mol kg-1 at >500 m (Fig. 3a and 4a, supplementary Figure 1). TDN concentrations increased with depth from 18±8 $\mu$mol l-1 and 22±7 $\mu$mol l-1 within the upper 20 m in April and June, respectively, and reached a maximum of 54 $\mu$mol l-1 at 850 m (Fig. 2c). DOC decreased with depth from 94 ±37 $\mu$mol l-1 and 69 ±12 $\mu$mol l-1 in the upper 20 m in April and June, respectively, to lowest values of 37 $\mu$mol l-1 at 850 m. The steepest gradient in DOC concentration was observed in the upper 20-60 m (Fig. 2d) during both cruises. "

AR1: Specific Comments.

Title: It does not reflect the measurements performed in the study. "Bacterial organic carbon uptake" was not measured.

AC: In the revised version we will change the title: "Bacterial degradation activity in the Eastern Tropical South Pacific oxygen minimum zone"

AR1: L19: Bacterial growth efficiency was taken from Rivkin and Legrende (2001) as a simple function of temperature. It should not be considered as a result from the present study.

In our study, we estimated Bacterial growth efficiency (BGE) by two independent methods as explained in chapter 2.5, line 157. One approximation includes the water temperature and uses the equation from Rivkin and Legendre (2001), the other is the

based on measured bacterial production and DOC loss rates. The BGE referred to within the abstract was calculated with latter method and is therefore a result of this study and independent from Rivkin and Legendre (2001). The results are described in section 3.3 and discussed in the paragraph starting in line 332.

AR1: L25: Gruber et al. is a good reference for global scale processes and future conditions, however, a better reference for the measurement of anoxic conditions in the ETNP OMZs would be: Tiano et al. 2014. Deep-Sea Res. Part I. 94, 173-183.

AC: Thank you, we will include this reference, in the revised version.

AR1: L28: One classical reference dealing with the extention and volumens of the different OMZs is Paulmier & Ruiz-Pino 2009. Progress in Oceanography 80, 113-128.

AC: Thank you, we will include this reference, in the revised version.

AR1: L36-37: DNRA might result in lower metabolic energy yield, but it is not a mayor pathway in OMZs. Although it has been found in the ETSP, it showed sporadic and low rates (Kalvelage et al. 2013). On the other hand, denitrification might be considered one of the main anaerobic heterotrophic process but it is yielding 99% of the energy compared to aerobic respiration, i.e. it is almost equally efficient. This paragraph seems to be biased to introduce the idea of inefficient anaerobic metabolism, but it is not proved.

AC: We will modify the paragraph and focus more on previous observations of reduced carbon fluxes in OMZs:

"Within OMZs, enhanced vertical carbon export has been observed (Devol and Hartnett, 2001; Roullier et al., 2014) and explained by potentially reduced microbial activity and consequently reduced organic matter remineralization in suboxic and anoxic waters. "

Further, we will add a reference showing that the energy yield gained by denitrifying bacteria is even less than suggested by the chemical equations: "Additionally, the en-
ergy yield available for the production of cell mass seems to be less than suggested by the chemical equations (Strohm et al., 2007)."

AR1: L51-58: The effect of oxygen concentration on bacterial production and extracellular enzymes activity was ambiguous before the comment of G.Taylor. When the differential particulate organic matter was considered, hydrolytic rates were similar. This paragraph needs then some rewording because the study is not clearly justified now.

AC: We will change the paragraph in the revised version:

"Investigations of hydrolysis rates as the initial step of organic matter degradation, may help to unravel possible adaptation strategies of bacterial communities to suboxic and anoxic conditions (Hoppe et al., 2002). For instance, high extracellular enzyme rates might compensate a putative lower energy yield of anaerobic respiration and the subsequent biogeochemical effects. However, very few studies have investigated the effect of oxygen on hydrolytic rates, so far. Hoppe et al. (1990) did not find differences between oxic and anoxic incubations of Baltic Sea water. In the Cariaco Basin, hydrolytic rates were significantly higher in oxic compared to anoxic waters (Taylor et al., 2009). However, this difference did not persist after rates were normalized to particulate organic matter concentration. The dependence of hydrolysis rates on organic matter concentrations described by Taylor et al. (2009), suggest an investigation of extracellular enzyme rates in a more productive oxygen depleted system. The Peruvian upwelling system is an ideal setting in this respect, as it allows to investigate extracellular enzyme activity at shallow oxyclines, with high amounts of labile organic matter (Loginova et al. 2019)."

AR1: L61-62: Again, I disagree with the "lower efficiency of anaerobic respiration" (unless other processes different than denitrification are proved to be relevant).

AC: The term "lower efficiency of anaerobic respiration" will be changed to "energy yield"

AR1: L86-87: It is not clear for me if the filter or the ampule were rinsed with the sample.

AC: In the revised version, we will clarify that the filter was rinsed with the sample. The ampules were combusted (500°C/ 8 h) and should not contain any organic carbon.

AR1: L96-106: To be consistent, what is the detection limit and precision of the DHAA and DCHO analysis?

AC: We will add this information: "Detection limit of DHAA was 1.4 nmol L-1 and 10 nmol L-1 for DCHO. The precision was 2% and 5% for DHAA and DCHO, respectively."

AR1: L116: Fig. 5 is cited before Fig. 2.

AC: This will be improved in the revised version.

AR1: L131: Bacterial Production was measured at 13_C for all samples. Considering the range of temperatures found along the water column (7-24 _C), incubation temperature was up to 12_C off the in situ temperature. There were no compensation for the temperature variation, probably leading to significant deviation from in situ estimates. Considering the relevance of these results for the discussion, authors should correct measured rates with in situ temperature.

AC: In the revised version of the manuscript, we will take in situ temperature into account when calculating bacterial production following the approach by López-Urrutia and Morán (2007). All calculations, figures and discussions throughout the text will be adapted.

AR1: L154: Enzymatic rates were also measured at a fixed temperature of 13_C. Could in situ temperature be taken into account?

AC: In the revised manuscript, we will apply a temperature correction for the extra-cellular enzyme rates to account for the differences between in situ and incubation temperature. The correction factor will be based on differences in extracellular enzyme rates after incubations at 22.4°C and 13°C at five stations during the cruises. All calculations, figures and discussions throughout the text will be adapted.

AR1: L159-160: Please improve the description of the "Gas tight incubator". Considering the oxygen concentration values in your "low oxygen" incubations (8-40 umol/kg), how realistic are the conclusions applied to the anoxic core from these incubations? Oxygen concentrations of 8 uM are way above the Km for microbial processes such as Oxygen respiration, ammonium and nitrite oxidation, for instance, and above the inhibition values for anammox and denitrification. Please, include in the discussion the possible limitation of the measurements considering the high oxygen values achieved in the incubations. AC: In the revised version we will provide further information in section 2.6: "For samples > 5 $\mu$mol O2 kg-1, in situ concentration, incubations were conducted under atmospheric oxygen conditions. Samples < 5 $\mu$mol O2 kg-1 in situ concentration were incubated in a gas tight incubator that had two openings to fill and flush with gas. For our experiments, the incubator was flushed and filled with N2 to reduce oxygen concentrations. Still, control measurements occasionally revealed oxygen concentrations of 8- 40 $\mu$mol O2 kg-1. Additionally, samples were in contact with oxygen during pipetting and measurement."

Further, we will include in the discussion that results have to be interpreted with care: " The extracellular enzymes rates of our study have to be interpreted carefully since incubation was not fully anoxic and the remaining oxygen might have biased the results. Still, we assume that most extracellular enzymes were present at the time of sampling and thus oxygen contamination during the incubations did not strongly influence the rate measurements."

AR1: L201 (and L314): TDN includes the inorganic fraction. Nitrate in OMZs increases with depth, and might reach values up to 30-40 uM (example: Lam et al. 2009. Proceedings of the National Academy of Sciences 106, 4752-4757), which might represent 80-100% of the measured TDN. Could the authors include inorganic nutrients and use DON instead?

AC: Because the fraction of DON in TDN is low compared to DIN, especially at depth, DON obtained by subtracting DIN from TDN has a relatively high error. Moreover, bacteria may also use DIN. We therefore think that for the purpose of this study, TDN is the more accurate value.

AR1: L261-270: It is not clear how the parameters (DOC loss) have been calculated, only ranges are shown and it feels like the ranges have been subtracted without including the apparent heterogeneity of the different stations. Based on the data shown in Fig. 5, the large differences in the oxyclines must result in large differences in diapycnal oxygen fluxes. Some separation in the data shown in Fig. 5 would be advisable. Anoxic conditions are reached at depths varying from 20 to 100 m, probably with very different values for the measured variables (Chl a, DOC: : :) too. Contrary, DOC values change quickly in the first 10 m, but seems to be relatively constant below. Dots are not connected with lines so it seems to be a pool of data without a clear pattern. All the station seems to be the same.

AC: We thank the reviewer for this advice. In the revised version, we will show DOC concentrations at the different stations by a line plot instead of a dotchart (Fig. 5). Since the DOC flux was calculated for each station separately, we accounted for differences between the stations (see section 2.3).

AR1: L275: DNRA might have lower energy yield, it is not so low for denitrification.

AC: We thank the reviewer for this comment. We found a study showing a lower energy yield of denitrification than expected (see answer to comment concerning 36-37). Still, we will justify our hypothesis of reduced bacterial activity within suboxic waters compared to the oxyclines by previous observations of reduced carbon fluxes in OMZ. In the revised manuscript, we will change this sentence to: "We expected reduced rates of organic matter degradation within oxygen depleted waters, since reduced bacterial degradation activity might explain enhanced carbon fluxes in suboxic and anoxic waters (Devol and Hartnett 2001)"

AR1: L291-292: I would delete "nitrous oxide" otherwise further explanation is needed as the contribution from anammox to N2O production is quite reduced.

AC: We thank the reviewer for this advice and will delete "nitrous oxide"

AR1: L296-297: Remove "respiratory" from "autotrophic anaerobic respiratory pathway". Babbin et al (2014, Science 344, 406-408) and Kalvelage et al. (2013. Nature Geosciences 6, 228-234) are also appropriate references for that quote. In addition, I would delete the sentence in L298-299, denitrification+anammox are included in the global estimations for N losses.

AC: We will include the references and will remove the word "respiratory". However, we do not understand the ambition to remove the sentence: "Our data indeed showed enhanced degradation of amino-acid-containing organic matter in low oxygen waters". This sentence does not indicate that denitrification and anammox are not included in the global estimation of N loss. It only indicates that our data are in line with the theory of high degradation of nitrogen compounds that might fuel anaerobic respiratory processes.

AR1: L301-307: This section exceed the results obtained in the present manuscript. A possible link to N cycle could be pointed, but the connection between hydrolysis and coupled denitrification-anammox is not supported.

AC: In the revised version of the manuscript, we will strictly separate the direct interpretation of our results and possible implications:

"...Thereby, a preferential degradation of amino acid containing organic matter in suboxic waters compared to oxic waters has been suggested (Van Mooy et al., 2002). Degradation of nitrogen compounds by heterotrophic bacteria (e.g. denitrifiers) in suboxic waters enables the release of ammonia and nitrite and subsequently may support anammox, an autotrophic anaerobic pathway (Babbin et al., 2014; Kalvelage et al., 2013; Lam and Kuypers, 2011; Ward, 2013). This interaction between denitrifiers and

anammox bacteria could fuel the loss of nitrogen to the atmosphere. Our data indeed showed enhanced degradation of amino-acid-containing organic matter in low oxygen waters. Indicators for protein decomposition, i.e. LAPase Vmax and the degradation rate of DHAA by LAPase, were more pronounced within the suboxic waters (Fig. 4b, d). Therefore, observed LAPase rates are in line with the hypothesis of preferred degradation of nitrogen compounds under suboxia. However, simultaneous rate measurements of protein hydrolysis, nitrate reduction (e.g. denitrification) and anammox are needed to prove an indirect stimulation of anammox by protein hydrolysis via denitrification. A close coupling between anammox and nitrate reducing bacteria has previously been shown for wastewater treatments. There, nitrate reducers directly take up organic matter excreted by the anammox bacteria which in turn benefit from the released nitrite by respiratory nitrate reduction (Lawson et al., 2017). In the Pacific, denitrifiers and anammox bacteria are separated in space and time (Dalsgaard et al., 2012), potentially weakening a direct inter-dependency. "

AR1: L314-316: Inorganic nitrogen might be the mayor fraction of TDN. This fact must be taken into account, especially if any stimulation of metabolism is considered.

AC: We would like to stick to TDN (see explanation above). However, we will include more detailed information about a possible contribution of TDN to cell growth and activity:

"While labile organic matter is decreasing with depth (e.g. Loginova et al. 2019), TDN (Figure 2c), especially inorganic nitrogen, is increasing with depth. Thus, high concentrations of inorganic nitrogen at the lower oxycline are available for heterotrophic and chemoautotrophic energy gains. For instance, the co-occurrence of nitrate reduction, that was still detected at 25 $\mu$mol O2 L-1, and microaerobic respiration might have stimulated cell-specific production or the accumulation of especially active bacterial species (Kalvelage et al. 2011, Kalvelage, 2015). "

AR1: L317-322 and L323-331: These paragraphs seem to be not finished. There are

no clear conclusion for the discussion of these results.

AC: We thank the reviewer for this remark and will finish the paragraphs with a concluding sentence in the revised version of the manuscript

" Depth distribution of cell-specific and total bacterial production was different (Fig. 3b, d); cell-specific production was reduced in suboxic waters, while total production was similar in suboxic waters compared to the oxycline. This suggests that lower cell-specific production was compensated by higher cell abundance within the suboxic waters (Figure 3c), resulting in an overall unhampered bacterial organic matter cycling in the OMZ core. One reason for the accumulation of cells within the OMZ might be reduced predation, suggesting the OMZ core as an ecological niche for slowly growing bacteria. Reduced grazing by bacterivores thus preserves bacterial biomass in suboxic waters from entering into the food chain. This way of bacterial biomass preservation has been suggested as possible explanation for enhanced carbon preservation in anoxic sediments by Lee (1992), and may also explain our observations for the anoxic water column.

" …. For instance, SAR406, SAR202, ACD39 and PAUC34f have the genetic potential for the turnover of complex carbohydrates and anaerobic respiratory processes, in the Gulf of Mexico (Thrash et al., 2017). Consequently, our findings of active bacterial degradation of DOM are supported by molecular biological studies. Still, simultaneous measurements of bacterial degradation and production have to be combined with molecular analysis in future studies off Peru."

AR1: L346-347: According to M&M, BGE followed the established temperature dependence. If no other parameter was used for its calculation, I cannot see how the results of this manuscript for this calculated (but not measured) parameter suggest that oxygen availability control bacterial growth efficiency.

AC: See answer for comment concerning line 19.

[Figure]

AR1: L365-367: Well, this study provides estimations, but does not provide measurements for carbon and oxygen losses.

AC: In the revised version of the manuscript, we will emphasize that we only can give estimates.

AR1: L378: Why a BGE of 20% is now assumed? BGE was estimated based on in situ temperature before.

AC: We thank the reviewer for this comment. First, we will explain the choice of a BGE of 20% in the revised manuscript: "The amount of carbon oxidized by denitrification based on the studies of Dalsgaard et al. (2012) and Kalvelage et al. (2013) can be converted into bacterial production applying a BGE. The average temperature dependent BGE was 20%. A BGE of 20% agrees well with other studies (Del Giorgio and Cole, 1998). Assuming a BGE of 20%, the denitrification rates in Dalsgaard et al. (2012) and Kalvelage et al. (2013) suggest a bacterial production of $\leq 5$ $\mu$mol C m-3 d-1, equivalent to only about 14% of total heterotrophic bacterial production in suboxic waters determined in our study."

In the revised version, we will also include a BGE of 6% into our calculations. For this we will not focus on the denitrification rates mentioned in the paragraph above, but on the sum of anaerobic carbon oxidation rates including denitrification, DNRA and simple nitrate reduction, as it is also discussed within the manuscript for a BGE of 20% (line 380). Absolut values will change within the revised version because of temperature correction: "The same calculation can be repeated assuming a BGE of 6%, which is the average BGE within this study based on DOC loss and bacterial production. Assuming a BGE of 6%, the estimated 109 $\mu$mol C m-3 d-1 that are respired by anaerobic carbon oxidation (Kalvelage et al., 2013) would represent 94% of the carbon uptake. Consequently, 7 $\mu$mol C m-3 d-1, i.e. 6% of the carbon uptake, are incorporated into the bacterial biomass. A bacterial biomass production of 7 $\mu$mol C m-3 d-1 is even lower than the bacterial production of 27 $\mu$mol C m-3 d-1, based on a BGE of 20% and cannot

explain the average bacterial production measured in suboxic waters during our study (37 $\mu$mol C m-3 d-1). Therefore, this estimation suggests higher rates of heterotrophic anaerobic respiratory processes than previously measured. Since denitrification rates were not measured directly, the comparability of published denitrification rates and our measurements of bacterial production are limited. However, our data suggest that the carbon oxidation potential off Peru is more evenly distributed than expected . . . .."

AR1: L383-390: The presented data for bacterial production can not be directly attributed to denitrification as it was not directly measured and the high oxygen levels during the BP measurements could have inhibited denitrification. The last and conclusive sentence seems to be pretentious.

AC: Samples of bacterial production were incubated in closed vials and bubbled with a N2/CO2 mixture (section 2.5). Therefore, we may assume ongoing anoxic respiratory processes such as denitrification. However, we will include the following sentence to account for the uncertainty (see also remark above): "Since denitrification rates were not measured directly, the comparability of published denitrification rates and our measurements of bacterial production are limited."

AR1: L392-400: Conclusions should be more attached to the obtained and proved results of the measurements. The measurements of bacterial production do not allow to prove the dominance of individual pathways and even less to link it with the production of nitrous oxide.

AC: We thank the reviewer and will delete the questionable part of the conclusion and instead refer to the search of alternative explanations for the enhanced carbon fluxes in OMZs compared to the oxygenated water (see also comment of the second reviewer line 399).

---

## Author Comment (AC3) · 13 Sep 2019

Anonymous Referee #2 (AR2): This is a useful study investigating the complicated microbial dynamics within oxygen minimum zones with many different biogeochemical and physical measurements made. The authors focus on calculating bacterial production predominantly associated with carbon cycling, but then also use other studies to consider the input of nitrogen cycling and anoxic processes. The manuscript is very well written, generally clear and detailed. I have a few suggestions to revise the text and figures to make some of the points clearer and to hopefully clarify some uncertainties. One point that was not mentionned was that the bacteria were collected from

suboxic concentrations but rates measured in oxic conditions I assume? How might the fact the microbes are being oxidised affect your results? It is difficult to work in OMZs and many of the studies cited would have done a similar thing but i think this should be discussed.

Author Comment (AC): We thank the reviewer for this comment. Regarding the extracellular enzyme rates, we are aware of having conducted a challenging method. This is because we had a trade-off between feasible measurements of extracellular enzyme rates at different substrate concentrations to calculate Vmax and reducing the contamination of oxygen. In the revised version, we will include in the discussion that results have to be interpreted with care: "The extracellular enzymes rates of our study have to be interpreted carefully since incubation was not fully anoxic and the remaining oxygen might have biased the results. Still, we assume that most extracellular enzymes were present at the time of sampling and thus oxygen contamination during the incubations did not strongly influence the rate measurements."

Samples of bacterial production were incubated in closed vials and bubbled with a N2/CO2 mixture (section 2.5)., avoiding oxygen contamination during the incubation time. Therefore, we assume that bacterial production was not affected by oxygenation.

AR2: Also the authors seemed to switch between top/bottom hypoxic and upper/lower oxycline, which i took to mean the same thing. If not this should be clarified.

AC: The term low_hypoxic is defined in the method section 2.7 (>5 to <20 $\mu$mol O2 kg-1). Therefore, it is correct that "bottom_low_hypoxic" is identical to the lower oxycline, since at the lower oxycline oxygen concentrations only increased up to 15 $\mu$mol. The term "upper_low_hypoxic" differs from the term "upper oxycline" since the upper oxycline includes waters with oxygen concentrations between 5 to 60 $\mu$mol L-1. Within the revised version of the manuscript we will replace the statistical test that were until now only done for the "upper_low_hypoxic" waters by statistical tests with samples from the entire oxycline, to be consistent.

AR2: Line 16 - Change to 'from the upper AND lower oxyclines', using 'or' makes it seem negative and I had to read it a few times to understand you were saying production was high.

AC: In the revised version, we will change the sentence to: "Nevertheless, high cell-specific bacterial production was observed in samples from oxyclines and extracellular enzyme rates were especially high at the lower oxycline, corroborating earlier findings of highly active and distinct micro-aerobic bacterial communities. "

AR2: Line 73 - I noticed the transects had data from both cruises. They are quite close together temporally, but even so some discussion on how the data is aggregated and if temporal affects are accounted for is needed. Did you look at the data separately per cruise too? Which data/transects are used in the figures? What is the seasonality like in the region?

AC: We thank the reviewer for his/her advice and agree with the proposal to include more information about the different cruises. Therefore, we will i) describe the seasonality within the sampling region in the introduction of the revised manuscript, ii) describe the study area in more detail for each cruise and iii) show the oxygen content, chlorophyll a concentrations and temperatures for each station in the revised supplement (see also first comment of the first reviewer). However, we prefer not to distinguish between cruises for statistical tests of bacterial production and extracellular enzyme rates since we focus on possible differences between oxygen regimes. Moreover, bacterial production was only sampled during April and combining extracellular enzyme rates of both cruises increases sampling size. Still in the revised manuscript we will include the represented cruises in the subtitle of the figures.

i) "In austral winter, upwelling and subsequently nutrient supply to surface waters strengthens (Bakun and Nelson 1991, Echevin et al 2008). However, Chl a concentration is highest in austral summer, with the seasonal amplitude being stronger for surface Chl a concentrations than for depth averaged Chl a concentrations (Echevin et al

2008). This is because in winter, phytoplankton growth is, next to iron, mainly limited by light due to deeper mixing , whereas in summer macronutrients are limiting phytoplankton growth (Echevin et al 2008). Further, irregular variability in the El Niño–Southern Oscillation raises (during La Niña) or deepens (El Niño) the upper oxycline. Subsequently, anaerobic respiratory processes become stimulated or restricted, respectively and biogeochemical cycling of organic and inorganic matter changes (Llanillo et al. 2013). During the year of sampling (2017), neither a strong La Niña nor an El Niño was detected (https://ggweather.com/enso/oni.htm). However, in January, February and March 2017 there was a strong coastal El Niño with enhanced warming (+1.5°C) of sea surface temperatures in the eastern Pacific (Gerreaud 2018)."

"During our two cruises to the Peruvian upwelling system (Fig. 1), seasonal variability caused higher maximum Chl a concentrations and warmer temperatures in April compared to June 2017. Chl a concentration reached up to 11 and 4 $\mu$g l-1 within the upper 25 m in April and June, respectively. Still, average Chl a concentrations within the upper 10 m (M136: 3.1±2.6 $\mu$g l-1; M138: 2.8 ± 1.3 $\mu$g l-1) were not significantly different between the two cruises (nM136=75, n M138= 40, W=1416, p= 0.6). At depths >50 m, Chl a concentration was generally below detection limit (Fig. 2a, supplementary Figure 1). Within the upper 10 m the water was warmer in April (21.3 ±1.6°C) compared to June (17.6 ± 0.6°C) (nM136=75, n M138= 40, W=2886, p<0.01) (Fig. 2b, supplementary Figure 1). Oxygen >100 $\mu$mol kg-1 was observed in the surface mixed layer, decreased steeply with depth and reached suboxic concentrations (<5$\mu$mol L-1) at > 60 ± 24 m (Fig. 3a and 4a, supplementary Figure 1). Shallowest depth with suboxic oxygen concentrations were 14 m in April (station Q) and 29 m in June (station D), probably influenced by the distance from shore (Q<D). Oxygen increased again to up to 15 $\mu$mol kg-1 at >500 m (Fig. 3a and 4a, supplementary Figure 1). TDN concentrations increased with depth from 18±8 $\mu$mol l-1 and 22±7 $\mu$mol l-1 within the upper 20 m in April and June, respectively, and reached a maximum of 54 $\mu$mol l-1 at 850 m (Fig. 2c). DOC decreased with depth from 94 ±37 $\mu$mol l-1 and 69 ±12 $\mu$mol l-1 in the upper 20 m in April and June, respectively, to lowest values of 37 $\mu$mol l-1 at 850 m.

The steepest gradient in DOC concentration was observed in the upper 20-60 m (Fig. 2d) during both cruises. "

AR2: Line 139 - citation for first use of BOD and write in full first time used in main text

AC: In the revised version, we will improve this sentence: "The bacterial oxygen demand (BOD) is the amount of oxygen needed to fully oxygenize organic carbon that has been taken up and not transformed into biomass by bacterial production (BP)"

AR2: Line 145 - what temperature dependance? Is a conversion used? Cite

AC: In the revised version, we will include the formula for temperature dependence of the cited study: "...ii) the bacterial growth efficiency (BGE) follows the established temperature dependence (BGE=0.374[±0.04] -0.0104[±0.002]T), resulting in a BGE between 0.1 and 0.3 in the depth range of 10-60 m and an in situ temperature of 14 to 19°C (Rivkin and Legendre, 2001) ..."

AR2: Line 198 - Perhaps place oxygen concentration in Fig. 2 with the other 'standard' oceanographic measurements, as it is a key plot for this paper. I would also add on horizontal layers onto the transect images, i.e. by using black lines to show oxycline/ omz/hypoxic/oxic layers.

AC: We appreciate the suggestion of the reviewer. However, we placed the oxygen concentration on purpose as first panel of Figure 2, since all statistical analysis are referred to this parameter and therefore it should appear together with the biological rates. This also enables an overview about the respective oxygen concentrations at sampling depth that we will further indicate by black lines in the revised version of the manuscript.

AR2: Also, what was lowest oxygen concentration, was it 5 umol/L so only suboxic or even lower to maybe anoxic as set by your definitions in the introduction? The study refers to 'suboxic' throughout which makes me think outside of the OMZ, but actually this is the OMZ. Being clear about this early on in the results would help.

AC: We thank the reviewer for this advice. Indeed, we only distinguished between suboxic and hypoxic conditions. We are confident that the OMZ core includes zones with oxygen concentrations below our detection limit, as it is described in section 3.1. This is also indicated by increased nitrate concentrations ($\sim$6$\mu$mol L-1) in the OMZ core, suggesting anaerobic reduction of nitrate (data not included in the manuscript). In the revised version we will include a sentence in section 3.1 to clarify which oxygen concentrations were relevant for our statistical analysis: "Oxygen decreased steeply with depth and fell below detection of Winkler titration. For further analysis and within the text in situ oxygen concentrations below 5 $\mu$mol O2 L-1 are referred to as "suboxic"."

AR2: Line 199 - Does this mean OMZ is 100-500 m depth, be explicit

AC: Within the revised version, we will be more explicit and reformulate this paragraph (see also comment concerning line 73) : "Oxygen >100 $\mu$mol kg-1 was observed in the surface mixed layer, decreased steeply with depth and reached suboxic concentrations (<5$\mu$mol L-1) at > 60 $\pm$ 24 m (Fig. 3a and 4a, supplementary Figure 1). Shallowest depth with suboxic oxygen concentrations were 14 m in April (station Q) and 29 m in June (station D), probably influenced by the distance from shore (Q<D). Oxygen increased again to up to 15 $\mu$mol kg-1 at >500 m (Fig. 3a and 4a, supplementary Figure 1)." Consequently, the suboxic waters was between 60 and 500 m. AR2: Line 206 - 'except for most coastal stations' - what happened at these stations? AC: In the revised version we will include an additional sentence (thereby bacterial production will differ from the submitted manuscript, since bacterial production will be corrected for differences between incubation and in situ temperature; see comments of first reviewer): "Bacterial production varied strongly throughout the study region and ranged from 0.2 to 2404 $\mu$mol C m-3 d-1 (Fig. 3b), decreased in general from surface to depth and showed significantly higher rates in the oxygenated surface compared to the OMZ (Fig. 3b). At the most coastal station (G) bacterial production remained high near the bottom depth of 75 m (280 $\mu$mol C m-2d-1 at 72 m) (Fig. 3b)"

AR2: Line 235 - full statistical results in parentheses is great to see and the correct way

to present results, however with so many tests and parts of the text in parentheses it stops the flow when reading. Can you shorten the statistical results in some way? Or add a table to the supplementary material?

AC: We agree, that the flow of reading is disturbed and will include the statistical results in the supplement.

AR2: Line 242 - normalisation completely changes the pattern of production with depth and oxygen, reverses it compared to un-normalised data. It would be good to show this and discuss further, perhaps using scatter plots too.

AC: We very much appreciate this comment of the reviewer. Producing the scatter plot helped us to see that the trends between cell-specific and total production are not completely invers. Still, cell-specific production is correlating more strongly with oxygen than total production. Thus, as described in the manuscript, cell abundance seems to counteract the lower cell-specific bacterial production at suboxic oxygen concentrations compared to the oxyclines. A further statistical test revealed that at the coastal stations (G and T) cell-specific production is more similar between suboxic waters and the oxycline. This suggests that the supply of organic matter stimulates bacterial production under suboxia. We will include this thought within the discussion of the revised manuscript: "Baltar et al. (2009) showed increasing cell-specific enzymatic rates and decreasing cell-specific bacterial production, with increasing depth in the subtropical Atlantic. Baltar et al. (2009) explained this pattern with decreasing organic matter lability. In our study, differences in cell-specific bacterial production between suboxic waters and the oxycline did not persist at the most coastal stations (G and T). This indicates the stimulation of bacterial activity, including anaerobic respiratory processes, by the input of labile organic matter from the shore. Therefore, our study suggests that a possible impairment of cell-specific bacterial production under suboxia is reduced by supply of organic matter. However, this hypothesis is restricted to a very limited number of samples and should be tested in further studies"

The scatter plot with cell-specific and total bacterial production rates in relation to oxygen will be included in the supplement of the revised manuscript and referred to within the results: "A detailed view at bacterial production in dependence of in-situ oxygen concentrations, reveals a stronger increase of cell-specific bacterial production, especially at < 10 $\mu$mol O2 L-1 at different stations (new supplementary Figure)."

AR2: Line 243 - Units of 'amol per cell per day' are incredibly low as one may expect from a 'per cell' measurement, but is this comparable with results from other studies?

AC: We agree with the reviewer that these rates seem low. Baltar et al. (2009) presents cell-specific production rates in the subtropical Atlantic (Figure 1 c) that varied between $\sim$ 0.006-0.03 fmol C cell d-1 (corresponding to 6-30 amol C cell d-1) between 96 and 503 m depth. Our original measurements range between 2-286 amol C cell d-1 between surface waters and $\sim$650 m depth. After temperature correction, cell-specific production rates ranged between 1 and 1120 amol C cell d-1. Consequently, our data include the measurement range of the former study in the Atlantic.

AR2: Line 284 - Is this finding because experiments were run in oxic conditions, as were some of the studies you cited too. But should consider the affects of exposing microbes from OMZ to oxygen.

AC: In the revised version, we will include in the discussion that results have to be interpreted with care (see also answer to the first comment of this reviewer): " The extracellular enzymes rates of our study have to be interpreted carefully since incubation was not fully anoxic and the remaining oxygen might have biased the results. Still, we assume that most extracellular enzymes were present at the time of sampling and thus oxygen contamination during the incubations did not strongly influence the rate measurements."

AR2: Line 375 - where did the data of amount of reduced nitrogen come from, was it Kalvelage? I found this section, whilst interesting, a little hard to follow which numbers were from this study and which from others. For instance, why use BGE from Del

Giorgio 1998 when you calculated it in this study?

AC: We thank the reviewer for this comment that includes one comment of the first reviewer. First, we will explain the choice of a BGE of 20% in the revised manuscript: "The amount of carbon oxidized by denitrification based on the studies of Dalsgaard et al. (2012) and Kalvelage et al. (2013) can be converted into bacterial production applying a BGE. The average temperature dependent BGE was 20%. A BGE of 20% agrees well with other studies (Del Giorgio and Cole, 1998). Assuming a BGE of 20%, the denitrification rates in Dalsgaard et al. (2012) and Kalvelage et al. (2013) suggest a bacterial production of ≤5 $\mu$mol C m-3 d-1, equivalent to only about 14% of total heterotrophic bacterial production in suboxic waters determined in our study."

Second, we will clearly indicate the source of data: "We compared bacterial production, i.e. rates of carbon incorporation, with denitrification rates previously reported for the South Pacific. Therefore, we converted one mol of reduced nitrogen that were measured by Dalsgaard et al. (2012) and Kalvelage et al. (2013) to 1.25 mol of oxidized carbon after the reaction equation given by Lam and Kuypers (2011). This calculation indicates that on average ≤19 $\mu$mol C m-3 d-1 are oxidized by denitrifying bacteria in the Eastern Tropical Pacific (Dalsgaard et al., 2012; Kalvelage et al., 2013). ….."

Third, we will expand the calculation and include a BGE of 6%. For this we will not focus on the denitrification rates mentioned in the paragraph above, but on the sum of anaerobic carbon oxidation rates including denitrification, DNRA and simple nitrate reduction, as it is also discussed within the manuscript for a BGE of 20% (line 380). Absolut values will change within the revised version because of temperature correction: "The same calculation can be repeated assuming a BGE of 6%, which is the average BGE within this study based on DOC loss and bacterial production. Assuming a BGE of 6%, the estimated 109 $\mu$mol C m-3 d-1 that are respired by anaerobic carbon oxidation (Kalvelage et al., 2013) would represent 94% of the carbon uptake. Consequently, 7 $\mu$mol C m-3 d-1, i.e. 6% of the carbon uptake, are incorporated into the bacterial biomass. A bacterial biomass production of 7 $\mu$mol C m-3 d-1 is even lower than the

bacterial production of 27 $\mu$mol C m-3 d-1, based on a BGE of 20% and cannot explain the average bacterial production measured in suboxic waters during our study (37 $\mu$mol C m-3 d-1). Therefore, this estimation suggests higher rates of heterotrophic anaerobic respiratory processes than previously measured. Since denitrification rates were not measured directly, the comparability of published denitrification rates and our measurements of bacterial production are limited. However, our data suggest that the carbon oxidation potential off Peru is more evenly distributed than expected ….."

AR2: Line 388 - Do you mean distributed evenly vertically or horizontally, or both?

AC: We were able to measure heterotrophic bacterial production at every depth and station. Therefore, we here suggest a more evenly horizontal and vertical distribution of heterotrophic anaerobic production, than indicated by heterotrophic anaerobic respiration measurements. We will add the words "horizontally " and "vertically", within the revised version.

AR2: Line 399 - I agree with final sentence of paper but not mentioned anywhere how can improve understanding or quantification processes, so on its own this final sentence is a bit weak for such a thorough study.

AC: We thank the reviewer and will add a suggestion for future studies in the conclusion of the revised manuscript:

"Our study suggests that suboxia does not reduce enzymatic degradation of organic matter and bacterial production in the Eastern Tropical South Pacific off Peru and therefore supports alternative explanations for the enhanced carbon export in OMZs compared to oxygenated waters. Differences between cell-specific and total rates of bacterial activity allude to different controls of cell abundance in suboxic systems and highlight the OMZ as a specific ecological niche. The combination of bacterial and physical rate measurements suggests that low BGEs in the upper oxycline contribute to sustaining the OMZ. Meanwhile, new findings during our study call for additional studies: i) DOC loss differed strongly between our investigation and the study of Loginova et al (2019). Therefore, combined physical and biological rate measurements in the Peruvian upwelling system should be repeated during austral summer, to learn more about the interplay of DOC loss and bacterial production rates during different seasons. ii) Integrated measurements of denitrification, microaerobic respiration and bacterial production are needed to estimate the fractions of incorporated and respired carbon under suboxia. The BGE received in that way could support or disprove the low BGE estimate, which was calculated from DOC loss and bacterial production in our study. Consequently, our study highlights the need for a better mechanistic understanding and quantification of processes responsible for oxygen and dissolved organic matter loss in OMZs that is inevitable to predict future patterns of deoxygenation in a warming climate."

AR2: Figures: Fig. 2 and Fig.3 - show horizontal oxygen regions as suggested above. Also, reduce extrapolation with ODV, large gap _ 100 km where no station/data between coastal and offshelf. Which interpolation did you use in ODV? The stations are running from the coast which is more east than offshore according to figure 1, so perhaps flip horizontally to reflect the east to west/coast to offshore nature of the spatial distribution. Having longitude instead of distance from coast (or both preferably) may be more useful, and make clear the inset is top 100 m.

AC: In the revised version, we will indicate the oxygen concentration at the sampling depths in Figure 2 and 3. Until now we used "Diva setting" with automatic scale adjustment, but in the revised version we will reduce the extrapolation to visualize the gap between stations. Further we will flip the y axis as well as indicate the longitude. Furthermore we will indicate the depth in the insets.

AR2: Fig 4 - labels of oxygen regimes different to text where instead oxyclines often referred to, is that the same as top and bottom hypoxic? Continuity throughout would be helpful.

AC: In the revised version, we will replace the words "top hypoxic" and "bottom hypoxic"
by "upper-" and "lower oxycline" in figure 4.

AR2: Fig. 5. Add a title for each panel so do not need to refer to legend as much.

AC: In the revised version, we will add subheadings to each panel of Figure 5.

———————————————————

[Figure]

[Figure]

**Fig. 1.** New supplementary Figure: Total vs cell-specific bacterial production with oxygen concentrations (left) and stations (right) indicated by color-coding

---

## Author Response (AR1)

**Answers to the first Referee**

**comments from referees/public:**

The manuscript by Maßmig et al. shows interesting results from two cruises in the ETSP OMZ off Peru. The combination of DOC, TDN, DHAA and DCHO with bacterial production and extracellular enzyme rates provides a nice overview of the microbial activity in general terms. Authors also show diapycnal fluxes for oxygen and DOC, including the potential role of microbial processes into those total fluxes. A similar manuscript has been recently published by the same authors (Loginova et al. 2019 Biogeosciences, 16). DOC, TDN, DON, DHAA, DCHO and diapycnal DOC and oxygen fluxes were also measured/estimated in a previous cruise in the same area. It is clear that the present study includes other data but discussion lacks a comparison between both studies and some results/conclusions seems to be repeated. For instance, the 33% of oxygen loss over depth attributed to bacterial oxygen demand is quite similar than in the previous study (38%). Please extend the discussion and comparison with the previous manuscript.

**author's response:**

We thank the reviewer for this comment and included a comparison between our study and Loginova et al. 2019 in the revised version of the manuscript. Additionally, we included a paragraph concerning the seasonality of the Peruvian system in the introduction (see comment of second Reviewer concerning line 73):

**author's changes in manuscript (page and line numbers refer to the revised manuscript pdf ):**

**page 14 line 445**

"Loginova et al. (2019) conducted similar physical rate measurements in the same study area with ~2 and ~10 times lower DOC and oxygen loss in the upper ~40 m compared to our study. Differences in loss rates were mainly caused by a ~ 10 times higher diapycnal diffusivity of mass in our study. This may have been caused by weaker stratification in the upper 100 m depth or differences in the turbulence conditions. Loginova et al. (2019) estimated a contribution of bacterial DOM degradation to oxygen loss (38 %) based on the loss of labile DOC (DHAA and DCHO). This value agrees well with our estimates of 18-33% of total oxygen loss, calculated under the assumption that DOC loss is solely attributed to bacterial degradation. However, the comparison of DOC and oxygen loss within each study revealed different patterns. Loginova et al. (2019) found a loss of DOC that clearly exceeded the loss of oxygen within the upper ~40 m. Hence, respiration of DOC could fully explain the observed oxygen loss in that study. In our study, more oxygen than DOC was lost over depth (Table 1). This loss of oxygen needs additional explanations such as degradation of particulate organic matter and physical mixing processes. One reason for the observed differences between the two studies that have been conducted in the same region might be seasonality. The study by Loginova et al. (2019) took place in austral summer, whereas our data were gained during austral winter. Water temperature was quite similar during both studies, probably due to the coastal El Niño one month before our sampling campaign (Garreaud, 2018). Still, the study by Loginova et al. (2019) included more stations with high Chl *a* concentrations (~8 µg L$^{-1}$), as typical for the austral summer, indicating a more productive system with more labile DOM (DCHO and DHAA). Prevalence of more labile DOM might explain the higher contribution of microbial DOM respiration to oxygen loss in the study by Loginova et al. (2019). Additionally, Loginova et al. (2019)

sampled with a much higher vertical resolution within the upper 140 m, restricting the direct comparability with our study. "

**comments from referees/public:**

The stations were sampled in two cruises (April and June) and distributed in three transects perpendicular to the coast: Lima, Paracas and Puerto Caballas (approx.). Spatial and temporal variability is however not considered in the manuscript. Some data correspond to some transects and cruise and other data correspond to other but no clear differentiation is included. Substances concentrations and fluxes were measured in Lima and Paracas transects in April, but enzymatic activity was measured in Paracas and Puerto Caballas in June. These data are however pooled and used for all the later estimations without any further consideration of spatiotemporal differences. Only one transect (Lima) is shown in Figures 2-3, are the conditions equal in the other transects (Temperature, Oxygen, Chlorophyl: : :)?

**author's response:**

With our approach we focus on possible differences between oxygen regimes. Hence, statistics of bacterial production and of extracellular enzyme rates are always related to the different oxygen concentrations. However, we included figures of oxygen and Chl *a* concentration and temperature for the remaining stations in the supplement (supplementary Fig. 1). Moreover, a more differentiated description of the study site and a comparison between cruises has been included in section 3.1:

**author's changes in manuscript:**

**page 8 line 244**

"During our two cruises to the Peruvian upwelling system (Fig. 1), maximum Chl *a* concentration was higher and temperatures were warmer in April compared to June 2017, probably representing seasonal variability. Chl *a* concentration reached up to 11 and 4 µg l$^{-1}$ within the upper 25 m in April and June, respectively. Still, average Chl *a* concentration at depth <10 m (M136: 3.1±2.6 µg l$^{-1}$; M138: 2.8 ± 1.3 µg l$^{-1}$) were not significantly different between the two cruises. At depths >50 m, Chl *a* concentration was generally below detection limit (Fig. 3a, supplementary Fig. 1). At depth < 10 m the water was warmer in April (21.3 ±1.6°C) than in June (17.6 ± 0.6°C) (Fig. 3b, supplementary Fig. 1). Oxygen concentration >100 µmol kg$^{-1}$ was observed in the surface mixed layer. Oxygen decreased steeply with depth, reached suboxic concentrations (<5 µmol kg$^{-1}$) at > 60 ± 24 m (Fig. 2c, 4a and 5a, supplementary Fig.1) and fell below detection of Winkler titration. For further analysis and within the text *in situ* oxygen concentrations <5 µmol O$_2$ kg$^{-1}$ are referred to as "suboxic". Shallowest depth with suboxic oxygen concentrations was 14 m in April (station Q) and 29 m in June (station D), probably representing that station Q was situated closer to the shore than station D. Oxygen increased again to up to 15 µmol kg$^{-1}$ at >500 m (Fig. 4a and 5a, supplementary Fig. 1). TDN concentrations increased with depth from 18±8 µmol l$^{-1}$ and 22±7 µmol l$^{-1}$ within the upper 20 m in April and June, respectively, and reached a maximum of 54 µmol l$^{-1}$ at 850 m (Fig. 3c). DOC decreased with depth from 94 ±37 µmol l$^{-1}$ and 69 ±12 µmol l$^{-1}$ in the upper 20 m in April and June, respectively, to lowest values of 37 µmol l$^{-1}$ at 850 m. The steepest gradient in DOC concentration was observed in the upper 20-60 m (Fig. 2b and 3d) during both cruises."

**comments from referees/public:**

Title: It does not reflect the measurements performed in the study. "Bacterial organic carbon uptake" was not measured.

**author's response:**

In the revised version we changed the title.

**author's changes in manuscript:**

**page 1 line 1**

"Bacterial degradation activity in the Eastern Tropical South Pacific oxygen minimum zone"

**comments from referees/public:**

L19: Bacterial growth efficiency was taken from Rivkin and Legrende (2001) as a simple function of temperature.

It should not be considered as a result from the present study.

**author's response:**

In our study, we estimated bacterial growth efficiency (BGE) by two independent methods as explained in chapter 2.5. One approximation includes the water temperature and uses the equation from Rivkin and

Legendre (2001), the other is based on measured bacterial production and DOC loss rates. The BGE

referred to within the abstract was calculated with latter method and is therefore a result of this study and independent from Rivkin and Legendre (2001). The results are described in section 3.3 and discussed in the discussion.

**comments from referees/public:**

L25: Gruber et al. is a good reference for global scale processes and future conditions, however, a better reference for the measurement of anoxic conditions in the ETNP OMZs would be: Tiano et al. 2014. Deep-Sea Res. Part I.

94, 173-183.

**author's response:**

Thank you, we included this reference, in the revised version.

**comments from referees/public:**

L28: One classical reference dealing with the extention and volumens of the different OMZs is Paulmier & Ruiz-

Pino 2009. Progress in Oceanography 80, 113-128.

**author's response:**

Thank you, we included this reference, in the revised version.

**comments from referees/public:**

L36-37: DNRA might result in lower metabolic energy yield, but it is not a mayor pathway in OMZs. Although it has been found in the ETSP, it showed sporadic and low rates (Kalvelage et al. 2013). On the other hand, denitrification might be considered one of the main anaerobic heterotrophic process but it is yielding 99% of the energy compared to aerobic respiration, i.e. it is almost equally efficient. This paragraph seems to be biased to introduce the idea of inefficient anaerobic metabolism, but it is not proved.

**author's response:**

We modified the paragraph and focus more on previous observations of reduced carbon fluxes in OMZs.

**author's changes in manuscript:**

**page 2 line 44**

"Within OMZs, enhanced vertical carbon export has been observed (Devol and Hartnett, 2001; Roullier et al., 2014) and explained by potentially reduced remineralization of organic matter in suboxic and anoxic waters."

**author's response:**

Further, we added a reference showing that the energy yield gained by denitrifying bacteria is even less than suggested by the chemical equations:

**author's changes in manuscript:**

**page 2 line 49**

"Additionally, the energy yield available for the production of cell mass seems to be less than suggested by the chemical equations (Strohm et al., 2007)."

**comments from referees/public:**

L51-58: The effect of oxygen concentration on bacterial production and extracellular enzymes activity was ambiguous before the comment of G.Taylor. When the differential particulate organic matter was considered, hydrolytic rates were similar. This paragraph needs then some rewording because the study is not clearly justified now.

**author's response:**

We changed the paragraph in the revised version:

**author's changes in manuscript:**

**page 3 line 66**

"Investigations of hydrolysis rates as the initial step of organic matter degradation, may help to unravel possible adaptation strategies of bacterial communities to suboxic and anoxic conditions (Hoppe et al.,

2002). High extracellular enzyme rates might compensate a putative lower energy yield of anaerobic respiration and the subsequent biogeochemical effects. However, very few studies have investigated the effect of oxygen on hydrolytic rates, so far. Hoppe et al. (1990) did not find differences between oxic and anoxic incubations of Baltic Sea water. In the Cariaco Basin, hydrolytic rates were significantly higher in oxic compared to anoxic water (Taylor et al., 2009). However, this difference did not persist after rates were normalized to particulate organic matter concentration. The dependence of hydrolysis rates on organic matter concentrations described by Taylor et al. (2009), suggest that productivity may play a role for extracellular enzymatic rates in oxygen depleted systems. The Peruvian upwelling system displays high amounts of labile organic matter (Loginova et al., 2019) at shallow oxyclines and thus allows for studying effects of low oxygen on extracellular enzyme rates under substrate replete conditions."

**comments from referees/public:**

L61-62: Again, I disagree with the "lower efficiency of anaerobic respiration" (unless other processes different than denitrification are proved to be relevant).

**author's response:**

The term "lower efficiency of anaerobic respiration" has been changed to "energy yield"

**comments from referees/public:**

L86-87: It is not clear for me if the filter or the ampule were rinsed with the sample.

**author's response:**

In the revised version, we clarified that the filter was rinsed with the sample. The ampules were combusted (500°C/ 8 h) and should not contain any organic carbon.

**author's changes in manuscript:**

**page 4 line 100**

"Samples were filtered through a syringe filter (0.45 μm glass microfiber GD/X membrane, Whatman

115 ™), that was rinsed with 50 ml sample, into a combusted glass ampoule (8 h, 500 °C)."

**comments from referees/public:**

L96-106: To be consistent, what is the detection limit and precision of the DHAA and DCHO analysis?

**author's response:**

We added this information:

**author's changes in manuscript:**

**page 5 line 119**

"Detection limit of DHAA was 1.4 nmol $L^{-1}$ depending on amino acid and 10 nmol $L^{-1}$ for DCHO. The precision was 2% and 5% for DHAA and DCHO, respectively."

**comments from referees/public:**

L116: Fig. 5 is cited before Fig. 2.

**author's response:**

This has been improved in the revised version.

**author's changes in manuscript:**

Figure order has been changed.

**comments from referees/public:**

L131: Bacterial Production was measured at 13°C for all samples. Considering the range of temperatures found along the water column (7-24 °C), incubation temperature was up to 12°C off the in situ temperature. There were no compensation for the temperature variation, probably leading to significant deviation from in situ estimates.

Considering the relevance of these results for the discussion, authors should correct measured rates with in situ temperature.

**author's response:**

In the revised version of the manuscript, we take *in situ* temperature into account when calculating bacterial production following the approach by López-Urrutia and Morán (2007). All calculations, figures and discussions throughout the text have been adapted.

**author's changes in manuscript:**

 **page 5 line 151**

"The incubation of samples at a constant temperature of 13°C resulted in deviations of max. 11°C between incubation ($T_{incubation}$) and *in situ* temperatures ($T_{insitu}$). In order to estimate *in situ* bacterial production from measured bacterial production during incubations, measured temperature differences were taken into account following the approach of López-Urrutia and Morán (2007). First, the temperature difference between $T_{insitu}$ and $T_{incubation}$ ($\delta T$) was computed in electron volt (ev$^{-1}$), after $T_{insitu}$ and $T_{incubation}$ (K) had been multiplied with the Boltzmann's constant *k (8.62x10$^{-5}$ eV K$^{-1}$):*

$\quad$ 1. $\quad \delta T\ [ev^{-1}] = \dfrac{1}{T_{incubation}[K]\ x\ k\ [evK^{-1}]} - \dfrac{1}{T_{insitu}[K]\ x\ k\ [evK^{-1}]}$

The decadal logarithm of *in situ* bacterial production ($\log_{10} BP_{insitu}$) was then calculated from the decadal logarithm of measured bacterial production during incubations ($\log_{10} BP_{incubation}$). Therefore we applied three different factors (*F*) depending on *in situ* Chl *a* concentration as proposed by López-Urrutia and

Morán (2007); with *F* being -0.583, -0.5 and -0.42 $[fgCcell^{-1}d^{-1}ev]$ for <0.5, 0.5-2 and >2 μg Chl *a* L$^{-}$

$^{1}$, respectively:

$\quad$ 2. $\quad log_{10}BP_{insitu}[fgCcell^{-1}d^{-1}] =$

$\qquad\qquad log_{10}BP_{incubation}[fgCcell^{-1}d^{-1}]\ +\ \delta T\ [ev^{-1}]x\ F\ [fgCcell^{-1}d^{-1}ev]$

Within the text, figures, equations and statistic results it is always referred to temperature corrected *in situ*

bacterial production. Temperature corrected bacterial production and original bacterial production measured during incubation can be compared in supplementary Table 2."

**comments from referees/public:**

L154: Enzymatic rates were also measured at a fixed temperature of 13°C. Could in situ temperature be taken into account?

$\quad$ **author's response:**

In the revised manuscript, we applied a temperature correction for the extracellular enzyme rates to account for the differences between *in situ* and incubation temperature. The correction factor was based on differences in extracellular enzyme rates after incubations at 22.4°C and 13°C at five stations during the cruises. All calculations, figures and discussions throughout the text were adapted.

$\quad$ **author's changes in manuscript:**

.**page 7 line 209**

"Similar to bacterial production, *in situ* extracellular enzyme rates were estimated based on extracellular enzyme rates measured during incubation. To account for the differences between *in situ* and incubation temperatures a correction factor (*F*) was applied based on differences in extracellular enzyme rates after additional incubations at 22.4°C next to the regular incubations at 13°C at five stations during the cruises.

The fluorescence signals at different substrate concentrations increased on average by a factor of 0.05 and

0.03 ($°C^{-1}$) for GLUCase and LAPase, respectively. Under the assumption that the increase in rates with temperature was linear, measured enzyme rates were adapted to *in situ* temperature, with (*EER$_{insitu}$; nmol*

$L^{-1} h^{-1}$) and ($EER_{incubation}$) being the *in situ* extracellular enzyme rates and extracellular enzyme rates during incubation, respectively:

1. $\delta T\ [°C] = T_{insitu}[°C] - T_{incubation}\ [°C]$

2. $EER_{insitu}[nmolL^{-1}h^{-1}] =$

$$EER_{incubation}[nmolL^{-1}h^{-1}] + EER_{incubation}[nmolL^{-1}h^{-1}]\ x\ F\ [°C^{-1}]\ x\ \delta T\ [°C]$$

Within the text, figures, equations and statistic results it is always referred to the temperature corrected *in situ* extracellular enzyme rates. Temperature corrected extracellular enzyme rates and original extracellular enzyme rates measured during incubation can be compared in supplementary Table 2."

**comments from referees/public:**

L159-160: Please improve the description of the "Gas tight incubator". Considering the oxygen concentration values in your "low oxygen" incubations (8-40 umol/kg), how realistic are the conclusions applied to the anoxic core from these incubations? Oxygen concentrations of 8 uM are way above the Km for microbial processes such as Oxygen respiration, ammonium and nitrite oxidation, for instance, and above the inhibition values for anammox and denitrification. Please, include in the discussion the possible limitation of the measurements considering the high oxygen values achieved in the incubations.

**author's response:**

In the revised version we provide further information in section 2.6:

**author's changes in manuscript:**

**page 6 line 188**

"For samples > 5 µmol *in situ* $O_2$ kg$^{-1}$ incubations were conducted under atmospheric oxygen conditions. Samples < 5 µmol *in situ* $O_2$ kg$^{-1}$ were incubated in a gas tight incubator that had two openings to fill and flush it with gas. For our experiment the incubator was flushed and filled with $N_2$, to reduce oxygen concentrations. Still control measurements occasionally revealed oxygen concentrations of 8 to 40 µmol $O_2$ kg$^{-1}$. Additionally, samples were in contact with oxygen during pipetting and measurement."

**author's response:**

Further, we included in the discussion that results have to be interpreted with care

**author's changes in manuscript:**

**page 11 line 344**

"The extracellular enzymes rates of our study have to be interpreted carefully since incubation was not fully anoxic and the remaining oxygen might have biased the results. Still, we assume that most extracellular enzymes were present at the time of sampling and thus oxygen contamination during the incubations did not strongly influence the rate measurements."

**comments from referees/public:**

L201 (and L314): TDN includes the inorganic fraction. Nitrate in OMZs increases with depth, and might reach values up to 30-40 uM (example: Lam et al. 2009. Proceedings of the National Academy of Sciences 106, 4752-

4757), which might represent 80-100% of the measured TDN. Could the authors include inorganic nutrients and use DON instead?

**author's response:**

Because the fraction of DON in TDN is low compared to DIN, especially at depth, DON obtained by subtracting DIN from TDN has a relatively high error. Moreover, bacteria may also use DIN. We therefore think that for the purpose of this study, TDN is the more accurate value (see also answer to comment on line 314).

**comments from referees/public:**

L261-270: It is not clear how the parameters (DOC loss) have been calculated, only ranges are shown and it feels like the ranges have been subtracted without including the apparent heterogeneity of the different stations. Based on the data shown in Fig. 5, the large differences in the oxyclines must result in large differences in diapycnal oxygen fluxes. Some separation in the data shown in Fig. 5 would be advisable. Anoxic conditions are reached at depths varying from 20 to 100 m, probably with very different values for the measured variables (Chl a, DOC: : :)

too. Contrary, DOC values change quickly in the first 10 m, but seems to be relatively constant below. Dots are not connected with lines so it seems to be a pool of data without a clear pattern. All the station seems to be the same.

**author's response:**

We thank the reviewer for this advice. In the revised version, we now show DOC concentrations at the different stations by a line plot instead of a dotchart (new Fig. 2). Since the DOC flux was calculated for each station separately, we accounted for differences between the stations (see section 2.3).

**comments from referees/public:**

L275: DNRA might have lower energy yield, it is not so low for denitrification.

**author's response:**

We thank the reviewer for this comment. We found a study showing a lower energy yield of denitrification than expected (see answer to comment concerning 36-37). Still, we justify our hypothesis of reduced bacterial activity within suboxic waters compared to the oxyclines by previous observations of reduced carbon fluxes in OMZ.

**author's changes in manuscript:**

**page 11 line 335**

"We expected reduced rates of organic matter degradation within oxygen depleted waters, since reduced bacterial degradation activity might explain enhanced carbon fluxes in suboxic and anoxic waters (Devol and Hartnett 2001)"

**comments from referees/public:**

L291-292: I would delete "nitrous oxide" otherwise further explanation is needed as the contribution from anammox to N2O production is quite reduced.

**author's response:**

We thank the reviewer for this advice and deleted "nitrous oxide"

**comments from referees/public:**

L296-297: Remove "respiratory" from "autotrophic anaerobic respiratory pathway". Babbin et al (2014, Science

344, 406-408) and Kalvelage et al. (2013. Nature Geosciences 6, 228-234) are also appropriate references for that quote. In addition, I would delete the sentence in L298-299, denitrification+anammox are included in the global estimations for N losses.

**author's response:**

We included the references and removed the word "respiratory". However, we do not understand the ambition to remove the sentence: "Our data indeed showed enhanced degradation of amino-acid- containing organic matter in low oxygen waters". This sentence does not indicate that denitrification anammox are not included in the global estimation of N loss. It only indicates that our data are in line with the theory of high degradation of nitrogen compounds that might fuel anaerobic respiratory processes.

**comments from referees/public:**

L301-307: This section exceed the results obtained in the present manuscript. A possible link to N cycle could be pointed, but the connection between hydrolysis and coupled denitrification-anammox is not supported.

**author's response:**

In the revised version of the manuscript, we strictly separate the direct interpretation of our results and possible implications:

**author's changes in manuscript:**

**page 11 line 358**

". . . Meanwhile, a preferential degradation of amino acid containing organic matter in suboxic waters compared to oxic waters has been suggested (Van Mooy et al., 2002). Degradation of nitrogen compounds by heterotrophic bacteria (e.g. denitrifiers) in suboxic waters enables the release of ammonia and nitrite and subsequently may support anammox, an autotrophic anaerobic pathway (Babbin et al., 2014;

Kalvelage et al., 2013; Lam and Kuypers, 2011; Ward, 2013). This interaction between denitrifiers and anammox bacteria could fuel the loss of nitrogen to the atmosphere. Our data indeed showed enhanced degradation of amino-acid-containing organic matter in low oxygen waters. Indicators for protein decomposition, i.e. LAPase $V_{max}$ and the degradation rate of DHAA by LAPase, were more pronounced within the suboxic waters (Fig. 5b, d). Therefore, observed LAPase rates are in line with the hypothesis of preferential degradation of nitrogen compounds under suboxia. However, simultaneous rate measurements of protein hydrolysis, nitrate reduction (e.g. denitrification) and anammox are needed to prove an indirect stimulation of anammox by protein hydrolysis via denitrification. A close coupling between anammox and nitrate reducing bacteria has previously been shown for wastewater treatments.

There, nitrate reducers directly take up organic matter excreted by the anammox bacteria which in turn benefit from the released nitrite by respiratory nitrate reduction (Lawson et al., 2017). In the Pacific, denitrifiers and anammox bacteria are separated in space and time (Dalsgaard et al., 2012), potentially weakening a direct inter-dependency."

**comments from referees/public:**

L314-316: Inorganic nitrogen might be the mayor fraction of TDN. This fact must be taken into account, especially if any stimulation of metabolism is considered.

 **author's response:**

 We would like to stick to TDN (see explanation above). However, we included more detailed information

 about a possible contribution of TDN to cell growth and activity:

 **author's changes in manuscript:**

 **page 12 line 383**

 "While labile organic matter is decreasing with depth (e.g. Loginova et al., 2019), TDN (Fig. 3c),

 especially inorganic nitrogen is increasing with depth. Thus, high concentrations of inorganic nitrogen at

 the lower oxycline are available for heterotrophic and chemoautotrophic energy gains. For instance, the

 co-occurrence of nitrate reduction, that was still detected at 25 $\mu$mol $O_2$ $L^{-1}$, and microaerobic respiration

 might have stimulated cell-specific production or the accumulation of especially active bacterial species

 (Kalvelage et al., 2011, 2015)."

**comments from referees/public:**

L317-322 and L323-331: These paragraphs seem to be not finished. There are no clear conclusion for the discussion of these results.

 **author's response:**

 We thank the reviewer for this remark and finished the paragraphs with a concluding sentence in the

 revised version of the manuscript

 **author's changes in manuscript:**

 **page 13 line 389**

 "Depth distribution of cell-specific and total bacterial production was different (Fig. 4b, d and

 supplementary Fig. 2); cell-specific production was significantly reduced in suboxic waters, while total

 production was more similar in suboxic waters compared to the oxycline. This suggests that lower cell-

 specific production was compensated by higher cell abundance within the suboxic waters (Fig. 4c),

 resulting in an overall unhampered bacterial organic matter cycling in the OMZ core. One reason for the

 accumulation of cells within the OMZ might be reduced predation, suggesting the OMZ core as an

 ecological niche for slowly growing bacteria. Reduced grazing by bacterivores thus preserves bacterial

 biomass in suboxic waters from entering into the food chain. This way of bacterial biomass preservation

 has been suggested as possible explanation for enhanced carbon preservation in anoxic sediments by Lee

 (1992), and may also explain our observations for the anoxic water column."

 **author's changes in manuscript:**

 **page 13 line 404**

" For instance, SAR406, SAR202, ACD39 and PAUC34f have the genetic potential for the turnover of complex carbohydrates and anaerobic respiratory processes, in the Gulf of Mexico (Thrash et al., 2017).

Consequently, our findings of active bacterial degradation of DOM are supported by molecular biological studies. Still, simultaneous measurements of bacterial degradation and production have to be combined with molecular analysis, in future studies off Peru.."

**comments from referees/public:**

L346-347: According to M&M, BGE followed the established temperature dependence. If no other parameter was used for its calculation, I cannot see how the results of this manuscript for this calculated (but not measured)

parameter suggest that oxygen availability control bacterial growth efficiency.

**author's response:**

Please see answer to comment concerning line 19.

**comments from referees/public:**

L365-367: Well, this study provides estimations, but does not provide measurements for carbon and oxygen losses.

**author's response:**

In the revised version of the manuscript, we emphasize that we only can give estimates.

**author's changes in manuscript:**

**page 14 line 443**

"…gives estimates for carbon and oxygen losses…"

**comments from referees/public:**

L378: Why a BGE of 20% is now assumed? BGE was estimated based on in situ temperature before.

**author's response:**

We thank the reviewer for this comment. First, we explain the choice of a BGE of 20% in the revised manuscript.

**author's changes in manuscript:**

**page 15 line 477**

" The amount of carbon oxidized by denitrification based on the studies of Dalsgaard et al. (2012) and

Kalvelage et al. (2013) can be converted into bacterial production applying a BGE. The average temperature dependent BGE was 20%. A BGE of 20% agrees well with other studies (Del Giorgio and

Cole, 1998). Assuming a BGE of 20%, the denitrification rates of Dalsgaard et al. (2012) and Kalvelage et al. (2013) suggest a bacterial production of $\leq 5$ µmol C m$^{-3}$ d$^{-1}$, equivalent to only about 14% of total average heterotrophic bacterial production in suboxic waters determined in our study."

**author's response:**

In the revised version, we also included calculations that are based on a BGE of 6%. For this we did not focus on the denitrification rates mentioned in the paragraph above, but on the sum of anaerobic carbon oxidation rates including denitrification, DNRA and simple nitrate reduction, as it is also discussed within the manuscript for a BGE of 20%. Absolut values changed within the revised version because of temperature correction:

**author's changes in manuscript:**

**page 15 line 489**

"The same calculation can be repeated assuming a BGE of 6%, which is the average BGE within this study based on DOC loss and bacterial production. Assuming a BGE of 6%, the estimated 109 µmol C $m^{-3}$ $d^{-1}$ that are respired by anaerobic carbon oxidation (Kalvelage et al., 2013) would represent 94% of the carbon uptake. Consequently, 7 µmol C $m^{-3}$ $d^{-1}$, i.e. 6% of the carbon uptake, are incorporated into the bacterial biomass. A bacterial biomass production of 7 µmol C $m^{-3}$ $d^{-1}$ is even lower than the bacterial production of 27 µmol C $m^{-3}$ $d^{-1}$, based on a BGE of 20% and cannot explain the average bacterial production measured in suboxic waters during our study (37 µmol C $m^{-3}$ $d^{-1}$). Therefore, this estimation suggests higher rates of heterotrophic anaerobic respiratory processes than previously measured. Since denitrification rates were not measured directly, the comparability of published denitrification rates and our measurements of bacterial production are limited. However, our data suggest that the carbon oxidation potential off Peru is more evenly horizontally and vertically distributed than expected ..."

**comments from referees/public:**

L383-390: The presented data for bacterial production can not be directly attributed to denitrification as it was not directly measured and the high oxygen levels during the BP measurements could have inhibited denitrification. The last and conclusive sentence seems to be pretentious.

**author's response:**

Samples of bacterial production were incubated in closed vials and bubbled with a N2/CO2 mixture (section 2.5). Therefore, we may assume ongoing anoxic respiratory processes such as denitrification. However, we included the following sentence to account for the uncertainty (see also remark above):

**author's changes in manuscript:**

**page 15 line 496**

"Since denitrification rates were not measured directly, the comparability of published denitrification rates and our measurements of bacterial production are limited."

**comments from referees/public:**

L392-400: Conclusions should be more attached to the obtained and proved results of the measurements. The measurements of bacterial production do not allow to prove the dominance of individual pathways and even less to link it with the production of nitrous oxide.

**author's response:**

We thank the reviewer and deleted the questionable part of the conclusion and instead refer to the search of alternative explanations for the enhanced carbon fluxes in OMZs compared to the oxygenated water (see also comment of the second reviewer line 399).

**Answers to the second Referee**

**comments from referees/public:**

This is a useful study investigating the complicated microbial dynamics within oxygen minimum zones with many different biogeochemical and physical measurements made. The authors focus on calculating bacterial production predominantly associated with carbon cycling, but then also use other studies to consider the input of nitrogen cycling and anoxic processes. The manuscript is very well written, generally clear and detailed. I have a few suggestions to revise the text and figures to make some of the points clearer and to hopefully clarify some uncertainties. One point that was not mentionned was that the bacteria were collected from suboxic concentrations but rates measured in oxic conditions I assume? How might the fact the microbes are being oxidised affect your results? It is difficult to work in OMZs and many of the studies cited would have done a similar thing but i think this should be discussed.

**author's response:**

We thank the reviewer for this comment. Regarding the extracellular enzyme rates, we are aware of having conducted a challenging method. This is because of the trade-off between feasible measurements of extracellular enzyme rates at different substrate concentrations to calculate Vmax and possible contamination of the sample with oxygen. In the revised version, we included an additional sentence to emphasize that results have to be interpreted with care.

**author's changes in manuscript:**

**page 11 line 344**

" The extracellular enzymes rates of our study have to be interpreted carefully since incubation was not fully anoxic and the remaining oxygen might have biased the results. Still, we assume that most extracellular enzymes were present at the time of sampling and thus oxygen contamination during the incubations did not strongly influence the rate measurements."

**author's response:**

Samples of bacterial production were incubated in closed vials and bubbled with a N2/CO2 mixture (section 2.5)., avoiding oxygen contamination during the incubation time. Therefore, we assume that bacterial production was not affected by oxygenation.

**comments from referees/public:**

Also the authors seemed to switch between top/bottom hypoxic and upper/lower oxycline, which i took to mean the same thing. If not this should be clarified.

**author's response:**

The term low_hypoxic is defined in the method section 2.7 ($>5$ to $<20$ $\mu$mol $O_2$ kg$^{-1}$). Therefore, it is correct that "bottom_low_hypoxic" is identical to the lower oxycline, since at the lower oxycline oxygen concentrations only increased up to 15 $\mu$mol. kg$^{-1}$ The term "upper_low_hypoxic" differs from the term "upper oxycline" since the upper oxycline includes waters with oxygen concentrations between 5 to 60 $\mu$mol kg$^{-1}$. Within the revised version of the manuscript we replaced the statistical test that were until now only done for the "upper_low_hypoxic" waters by statistical tests with samples from the entire oxycline, to be consistent.

**comments from referees/public:**

Line 16 - Change to 'from the upper AND lower oxyclines', using 'or' makes it seem negative and I had to read
it a few times to understand you were saying production was high.

    **author's response:**

    In the revised version, we changed the sentence.

    **author's changes in manuscript:**

    **page 1 line 15**

    "Nevertheless, high cell-specific bacterial production was observed in samples from oxyclines and cell-
    specific extracellular enzyme rates were especially high at the lower oxycline, corroborating earlier
    findings of highly active and distinct micro-aerobic bacterial communities. "

**comments from referees/public:**

Line 73 - I noticed the transects had data from both cruises. They are quite close together temporally, but even so
some discussion on how the data is aggregated and if temporal affects are accounted for is needed. Did you look
at the data separately per cruise too? Which data/transects are used in the figures? What is the seasonality like in
the region?

    **author's response:**

    We thank the reviewer for his/her advice and agree with the proposal to include more information about
    the different cruises. Therefore, we i) describe the seasonality within the sampling region in the
    introduction of the revised manuscript, ii) describe the study area in more detail for each cruise and iii)
    show the oxygen content, chlorophyll $a$ concentrations and temperatures for each station in the revised
    supplement (see also first comment of the first reviewer). However, we prefer not to distinguish between
    cruises for statistical tests of bacterial production and extracellular enzyme rates, since we focus on
    possible differences between oxygen regimes. Moreover, bacterial production was only sampled during
    April and combining extracellular enzyme rates of both cruises increases sampling size. Still, in the
    revised manuscript we included the represented cruises in the subtitle of the figures.

    **author's changes in manuscript:**

    **page 2 line 34**

    i) "In austral winter, upwelling and subsequently the nutrient supply to the surface waters increase
    (Bakund and Nelson, 1991; Echevin et al., 2008). However, chlorophyll $a$ (Chl $a$) concentration is highest
    in austral summer, with the seasonal amplitude being stronger for surface than for depth averaged Chl $a$
    concentrations (Echevin et al., 2008). In winter, phytoplankton growth is, next to iron, mainly limited by
    light due to the deeper mixing, whereas in summer macronutrients can become a limiting factor (Echevin
    et al., 2008). Further, El Niño–Southern Oscillation may affect organic matter cycling in the area since it
    affects the depth of the oxycline and therefore the extent of anaerobic processes in the upper water column
    (Llanillo et al., 2013). During the year of this study (2017), neither a strong La Niña nor a strong El Niño
    was detected (https://ggweather.com/enso/oni.htm). However, in January, February and March 2017 there
    was a strong coastal El Niño with enhanced warming (+1.5°C) of sea surface temperatures in the eastern
    Pacific (Garreaud, 2018)."

    **author's changes in manuscript:**

**page 8 line 244**

"During our two cruises to the Peruvian upwelling system (Fig. 1), maximum Chl *a* concentration was higher and temperatures were warmer in April compared to June 2017, probably representing seasonal variability. Chl *a* concentration reached up to 11 and 4 µg l$^{-1}$ within the upper 25 m in April and June, respectively. Still, average Chl *a* concentration at depth <10 m (M136: 3.1±2.6 µg l$^{-1}$; M138: 2.8 ± 1.3 µg l$^{-1}$) were not significantly different between the two cruises. At depths >50 m, Chl *a* concentration was generally below detection limit (Fig. 3a, supplementary Fig. 1). At depth < 10 m the water was warmer in April (21.3 ±1.6°C) than in June (17.6 ± 0.6°C) (Fig. 3b, supplementary Fig. 1). Oxygen concentration >100 µmol kg$^{-1}$ was observed in the surface mixed layer. Oxygen decreased steeply with depth, reached suboxic concentrations (<5 µmol kg$^{-1}$) at > 60 ± 24 m (Fig. 2c, 4a and 5a, supplementary Fig.1) and fell below detection of Winkler titration. For further analysis and within the text *in situ* oxygen concentrations <5 µmol O$_2$ kg$^{-1}$ are referred to as "suboxic". Shallowest depth with suboxic oxygen concentrations was 14 m in April (station Q) and 29 m in June (station D), probably representing that station Q was situated closer to the shore than station D. Oxygen increased again to up to 15 µmol kg$^{-1}$ at >500 m (Fig. 4a and 5a, supplementary Fig. 1). TDN concentrations increased with depth from 18±8 µmol l$^{-1}$ and 22±7 µmol l$^{-1}$ within the upper 20 m in April and June, respectively, and reached a maximum of 54 µmol l$^{-1}$ at 850 m (Fig. 3c). DOC decreased with depth from 94 ±37 µmol l$^{-1}$ and 69 ±12 µmol l$^{-1}$ in the upper 20 m in April and June, respectively, to lowest values of 37 µmol l$^{-1}$ at 850 m. The steepest gradient in DOC concentration was observed in the upper 20-60 m (Fig. 2b and 3d) during both cruises."

**comments from referees/public:**

Line 139 - citation for first use of BOD and write in full first time used in main text

**author's response:**

In the revised version, we improved this sentence.

**author's changes in manuscript:**

**page 6 line 168**

"The bacterial oxygen demand (BOD; mmol O$_2$ m$^{-3}$ d$^{-1}$) is the amount of oxygen needed to fully oxygenize organic carbon that has been taken up and not transformed into biomass by bacterial production (mmol C m$^{-3}$ d$^{-1}$)."

**comments from referees/public:**

Line 145 - what temperature dependance? Is a conversion used? Cite

**author's response:**

In the revised version, we included the formula for temperature dependence of the cited study:

**author's changes in manuscript:**

**page 6 line 175**

". . . ii) the bacterial growth efficiency (BGE) follows the established temperature dependence (BGE=0.374[±0.04] -0.0104[±0.002]T [°C]), resulting in a BGE between 0.1 and 0.3 in the depth range of 10-60 m and an *in situ* temperature of 14 to 19°C (Rivkin and Legendre, 2001)."

**comments from referees/public:**

Line 198 - Perhaps place oxygen concentration in Fig. 2 with the other 'standard' oceanographic measurements, as it is a key plot for this paper. I would also add on horizontal layers onto the transect images, i.e. by using black lines to show oxycline/ omz/hypoxic/oxic layers.

**author's response:**

We appreciate the suggestion of the reviewer. However, we placed the oxygen concentration on purpose as first panel of the new Figure 4, since all statistical analysis are referred to this parameter and therefore it should appear together with the biological rates. This also enables an overview about the respective oxygen concentrations at sampling depth that we further indicated by black lines in the revised version of the manuscript.

**comments from referees/public:**

Also, what was lowest oxygen concentration, was it 5 umol/L so only suboxic or even lower to maybe anoxic as set by your definitions in the introduction? The study refers to 'suboxic' throughout which makes me think outside of the OMZ, but actually this is the OMZ. Being clear about this early on in the results would help.

**author's response:**

We thank the reviewer for this advice. Indeed, we only distinguished between suboxic and hypoxic conditions. We are confident that the OMZ core includes zones with oxygen concentrations below our detection limit, as it is described in section 3.1. This is also indicated by increased nitrate concentrations ($\sim 6 \mu mol\ L^{-1}$) in the OMZ core, suggesting anaerobic reduction of nitrate (data not included in the manuscript). In the revised version we included a sentence in section 3.1 to clarify which oxygen concentrations were relevant for our statistical analysis.

**author's changes in manuscript:**

**page 8 line 250**

"Oxygen decreased steeply with depth, reached suboxic concentrations ($<5\ \mu mol\ kg^{-1}$) at $> 60 \pm 24$ m (Fig. 2c, 4a and 5a, supplementary Fig.1) and fell below detection of Winkler titration. For further analysis and within the text *in situ* oxygen concentrations $<5\ \mu mol\ O_2\ kg^{-1}$ are referred to as "suboxic"."

**comments from referees/public:**

Line 199 - Does this mean OMZ is 100-500 m depth, be explicit

**author's response:**

Within the revised version, we are more explicit and reformulate this paragraph (see also comment concerning line 73) :

**author's changes in manuscript:**

**page 8 line 250**

"Oxygen decreased steeply with depth, reached suboxic concentrations ($<5\ \mu mol\ kg^{-1}$) at $> 60 \pm 24$ m (Fig. 2c, 4a and 5a, supplementary Fig.1) and fell below detection of Winkler titration. For further analysis and within the text *in situ* oxygen concentrations $<5\ \mu mol\ O_2\ kg^{-1}$ are referred to as "suboxic". Shallowest depth with suboxic oxygen concentrations was 14 m in April (station Q) and 29 m in June (station D), probably representing that station Q was situated closer to the shore than station D. Oxygen increased again to up to 15 $\mu mol\ kg^{-1}$ at $>500$ m (Fig. 4a and 5a, supplementary Fig. 1)."

**author's response:**

Consequently, the suboxic waters was between 60 and 500 m.

**comments from referees/public:**

Line 206 - 'except for most coastal stations' - what happened at these stations?

**author's response:**

In the revised version we included an additional sentence (bacterial production differs from the submitted manuscript, since bacterial production was corrected for differences between incubation and in situ temperature; see comments of first reviewer):

**author's changes in manuscript:**

**page 9 line 262**

"Bacterial production varied strongly throughout the study region and ranged from 0.2 to 2404 µmol C

$m^{-3}$ $d^{-1}$ (Fig. 4b), decreased in general from surface to depth (except for the most coastal station) and showed significantly higher rates in the oxygenated surface compared to the OMZ (Fig. 4b). At the most coastal station (G) bacterial production remained high near the bottom depth of 75 m (280 µmol C $m^{-3}$ $d^{-}$

$^{1}$ at 72 m) (Fig. 4b)."

**comments from referees/public:**

Line 235 - full statistical results in parentheses is great to see and the correct way to present results, however with so many tests and parts of the text in parentheses it stops the flow when reading. Can you shorten the statistical results in some way? Or add a table to the supplementary material?

**author's response:**

We agree, that the flow of reading is disturbed and included the statistical results in the supplement.

**author's changes in manuscript:**

see new supplementary Table 2

**comments from referees/public:**

Line 242 - normalisation completely changes the pattern of production with depth and oxygen, reverses it compared to un-normalised data. It would be good to show this and discuss further, perhaps using scatter plots too.

**author's response:**

We very much appreciate this comment of the reviewer. Producing the scatter plot helped us to see that the trends between cell-specific and total production are not completely invers. Still, cell-specific production is correlating more strongly with oxygen than total production. Thus, as described in the manuscript, cell abundance seems to counteract the lower cell-specific bacterial production at suboxic oxygen concentrations compared to the oxyclines. A further statistical test revealed that cell-specific production is more similar between suboxic waters and the oxycline, at the coastal stations (G and T).

This suggests that the supply of organic matter stimulates bacterial production under suboxia. We included this thought within the discussion of the revised manuscript.

**author's changes in manuscript:**

**page 12 line 376**

"Baltar et al. (2009) showed increasing cell-specific enzymatic rates and decreasing cell-specific bacterial production, with increasing depth in the subtropical Atlantic and related this pattern to decreasing organic matter lability. In our study, differences in cell-specific bacterial production between suboxic waters and the oxycline did not persist at the most coastal stations (G and T). This indicates the stimulation of bacterial activity, including anaerobic respiratory processes, by the high input of labile organic matter. Therefore, our study suggests that a possible impairment of cell-specific bacterial production under suboxia is reduced by supply of organic matter. However, this hypothesis is restricted to a very limited number of samples and should be tested in further studies."

**author's response:**

The scatter plot with cell-specific and total bacterial production rates in relation to oxygen is included in the supplement of the revised manuscript and referred to within the results.

**author's changes in manuscript:**

**page 10 line 305**

"A detailed view at total- and cell-specific bacterial production in dependence of *in-situ* oxygen concentrations, reveals a stronger increase of cell-specific bacterial production, especially at <10 µmol $O_2$ kg$^{-1}$ at different stations (supplementary Fig. 2)."

**comments from referees/public:**

Line 243 - Units of 'amol per cell per day' are incredibly low as one may expect from a 'per cell' measurement, but is this comparable with results from other studies?

**author's response:**

We agree with the reviewer that these rates seem low. Baltar et al. (2009) presents cell-specific production rates in the subtropical Atlantic (Figure 1 c) that varied between ~0.006-0.03 fmol C cell d$^{-1}$ (corresponding to 6-30 amol C cell d$^{-1}$) between 96 and 503 m depth. Our original measurements ranged between 2-286 amol C cell d$^{-1}$ between surface waters and ~650 m depth. After temperature correction, cell-specific production rates ranged between 1 and 1120 amol C cell d$^{-1}$. Consequently, our data include the measurement range of the former study in the Atlantic.

**comments from referees/public:**

Line 284 - Is this finding because experiments were run in oxic conditions, as were some of the studies you cited too. But should consider the affects of exposing microbes from OMZ to oxygen.

**author's response:**

In the revised version, we included in the discussion that results have to be interpreted with care (see also answer to the first comment of this reviewer).

**author's changes in manuscript:**

**page 11 line 344**

"The extracellular enzymes rates of our study have to be interpreted carefully since incubation was not fully anoxic and the remaining oxygen might have biased the results. Still, we assume that most extracellular enzymes were present at the time of sampling and thus oxygen contamination during the incubations did not strongly influence the rate measurements."

**comments from referees/public:**

Line 375 - where did the data of amount of reduced nitrogen come from, was it Kalvelage? I found this section, whilst interesting, a little hard to follow which numbers were from this study and which from others. For instance, why use BGE from Del Giorgio 1998 when you calculated it in this study?

**author's response:**

We thank the reviewer for this comment that includes one comment of the first reviewer. First, we explain the choice of a BGE of 20% in the revised manuscript:

**author's changes in manuscript:**

**page 15 line 477**

" The amount of carbon oxidized by denitrification based on the studies of Dalsgaard et al. (2012) and Kalvelage et al. (2013) can be converted into bacterial production applying a BGE. The average temperature dependent BGE was 20%. A BGE of 20% agrees well with other studies (Del Giorgio and Cole, 1998). Assuming a BGE of 20%, the denitrification rates of Dalsgaard et al. (2012) and Kalvelage et al. (2013) suggest a bacterial production of $\leq 5$ µmol C m$^{-3}$ d$^{-1}$, equivalent to only about 14% of total average heterotrophic bacterial production in suboxic waters determined in our study."

**author's response:**

Second, we clearly indicated the source of data:

**author's changes in manuscript:**

**page 14 line 472**

"We compared bacterial production, i.e. rates of carbon incorporation, with denitrification rates previously reported for the South Pacific. Therefore, we converted one mol of reduced nitrogen that were measured by Dalsgaard et al. (2012) and Kalvelage et al. (2013) to 1.25 mol of oxidized carbon after the reaction equation given by Lam and Kuypers (2011). This calculation indicates that on average $\leq 19$ µmol C m$^{-3}$ d$^{-1}$ are oxidized by denitrifying bacteria in the Eastern Tropical Pacific (Dalsgaard et al., 2012; Kalvelage et al., 2013)....."

**author's response:**

Third, we expanded the calculation and include a BGE of 6%. For this we did not focus on the denitrification rates mentioned in the paragraph above, but on the sum of anaerobic carbon oxidation rates including denitrification, DNRA and simple nitrate reduction, as it is also discussed within the manuscript for a BGE of 20% (line 380). Absolut values changed within the revised version because of temperature correction.

**author's changes in manuscript:**

**page 15 line 489**

"The same calculation can be repeated assuming a BGE of 6%, which is the average BGE within this study based on DOC loss and bacterial production. Assuming a BGE of 6%, the estimated 109 µmol C m$^{-3}$ d$^{-1}$ that are respired by anaerobic carbon oxidation (Kalvelage et al., 2013) would represent 94% of the carbon uptake. Consequently, 7 µmol C m$^{-3}$ d$^{-1}$, i.e. 6% of the carbon uptake, are incorporated into the bacterial biomass. A bacterial biomass production of 7 µmol C m$^{-3}$ d$^{-1}$ is even lower than the bacterial production of 27 µmol C m$^{-3}$ d$^{-1}$, based on a BGE of 20% and cannot explain the average bacterial production measured in suboxic waters during our study (37 µmol C m$^{-3}$ d$^{-1}$). Therefore, this estimation suggests higher rates of heterotrophic anaerobic respiratory processes than previously measured. Since denitrification rates were not measured directly, the comparability of published denitrification rates and our measurements of bacterial production are limited. However, our data suggest that the carbon oxidation potential off Peru is more evenly horizontally and vertically distributed than expected ..."

**comments from referees/public:**

Line 388 - Do you mean distributed evenly vertically or horizontally, or both?

**author's response:**

We were able to measure heterotrophic bacterial production at every depth and station. Therefore, we here suggest a more evenly horizontal and vertical distribution of heterotrophic anaerobic production, than indicated by heterotrophic anaerobic respiration measurements. We added the words "horizontally " and "vertically", within the revised version.

**comments from referees/public:**

Line 399 - I agree with final sentence of paper but not mentioned anywhere how can improve understanding or quantification processes, so on its own this final sentence is a bit weak for such a thorough study.

**author's response:**

We thank the reviewer and added suggestions for future studies in the conclusion of the revised manuscript.

**author's changes in manuscript:**

**page 16 line 501**

"Our study suggests that suboxia does not reduce bacterial degradation of organic matter in the Eastern Tropical South Pacific off Peru. Bacterial species are seemingly adapted to these environments and higher cell abundance compensates for hampered cell-specific bacterial production under suboxia. Therefore, the previously observed enhanced carbon export in OMZs compared to oxygenated waters requires alternative explanations. Differences between cell-specific and total rates of bacterial activity allude to different controls of cell abundance in suboxic systems, highlighting the OMZ as a specific ecological niche. The combination of bacterial and physical rate measurements suggests that low BGEs in the upper oxycline contribute to sustaining the OMZ. Meanwhile, new findings during our study call for additional studies: i) DOC loss differed strongly between our investigation and the study of Loginova et al. (2019). Therefore, combined physical and biological rate measurements in the Peruvian upwelling system should be repeated during austral summer, to learn more about the interplay of DOC loss and bacterial production during different seasons. ii) Integrated measurements of denitrification, microaerobic respiration and bacterial production are needed to estimate the fractions of incorporated and respired carbon under suboxia. The BGE received in that way could support or disprove the low BGE estimate, which was calculated from DOC loss and bacterial production in our study. Consequently, our study highlights the need for a better mechanistic understanding and quantification of processes responsible for oxygen and DOM loss in OMZs that is inevitable to predict future patterns of deoxygenation in a warming climate."

**comments from referees/public:**

Figures: Fig. 2 and Fig.3 - show horizontal oxygen regions as suggested above. Also, reduce
extrapolation with ODV, large gap _ 100 km where no station/data between coastal and offshelf. Which
interpolation did you use in ODV? The stations are running from the coast which is more east than
offshore according to figure 1, so perhaps flip horizontally to reflect the east to west/coast to offshore
nature of the spatial distribution. Having longitude instead of distance from coast (or both preferably)
may be more useful, and make clear the inset is top 100 m.

**author's response:**
In the revised version, we indicated the oxygen concentration at the sampling depths in new Figure 3 and
4. Until now we used "Diva setting" with automatic scale adjustment, but in the revised version we
reduced the extrapolation to visualize the gap between stations. Further we flipped the y axis as well as
indicate the longitude. Furthermore, we indicated the depth in the insets.

**author's changes in manuscript:**
see new Figures 3 and 4

**comments from referees/public:**
Fig 4 - labels of oxygen regimes different to text where instead oxyclines often referred to, is that the same as top
and bottom hypoxic? Continuity throughout would be helpful.

**author's response:**
In the revised version, we replaced the words "top hypoxic" and "bottom hypoxic" by "upper-" and
"lower oxycline" in new Figure 5.

**author's changes in manuscript:**
see new Figure 5

**comments from referees/public:**
Fig. 5. Add a title for each panel so do not need to refer to legend as much.

**author's response:**
In the revised version, we added subheadings to each panel of new Figure 2.

**author's changes in manuscript:**
see new Figure 2

[revised manuscript text omitted]

Figure 5

---

## Author Response (AR2)

**Answers to the Referee**

**comments from referees/public:**

L462-463: The higher vertical resolution of the previous study (Loginova et al. 2019) does not exclude a direct comparison with the present one, with lower vertical resolution. Why is not possible? I could understand it the other way around.

> **author's response:**
>
> We thank the reviewer for this comment and agree that a comparison between the loss rates based on a higher vertical sampling frequency (Loginova et al. 2019) and our dataset is possible. Our statement arose because a higher vertical resolution might have resulted in slightly different loss rates within our study. However, we will delete the questionable sentence and restrict the discussion to the different seasons and the labile DOM concentrations that are mentioned within the same paragraph.
>
> **author's changes in manuscript (page and line numbers refer to the revised manuscript pdf ):**
>
> **page 14 line 462**
>
> The sentence "Additionally, Loginova et al. (2019) sampled with a much higher vertical resolution within the upper 140 m, restricting the direct comparability with our study." has been deleted.

**comments from referees/public:**

L465-470: Annamox has been proved to depend of the ammonium production by denitrification in the anoxic core (Babbin et al. 2014. Science 344, 406-408; Ward 2013. Science 341, 352-353.) and, due to the stoichiometry of the OM, only represent about 25-30% of the total rates. The presence of heterotrophic metabolism does not exclusively imply a dominance of denitrification as other processes (such as sulfate reduction, Candfield et al. 2010. Science 330, 1375-1378) might be equally relevant. It might be OK to later assume than the estimated rates are caused by denitrification, but this study does not provide clear evidences to show "the widespread occurrence of heterotrophic denitrification processes in the Peruvian OMZ".

> **author's response:**
>
> We are grateful for this comment and included a possible contribution of sulfate reduction to heterotrophic anaerobic processes within this paragraph. We further mention the general dominance of denitrification in relation to anammox, based on the stoichiometry of organic matter
>
> **author's changes in manuscript (page and line numbers refer to the revised manuscript pdf ):**
>
> **page 14 line 466**
>

[revised manuscript text omitted]